# DevBench: A Realistic, Developer-Informed Benchmark for Code Generation Models

## Abstract

DevBench is a telemetry-driven benchmark designed to evaluate Large Language Models (LLMs) on realistic code completion tasks. It includes 1,800 evaluation instances across six programming languages and six task categories derived from real developer telemetry, such as API usage and code purpose understanding. Unlike prior benchmarks, it emphasizes ecological validity, avoids training data contamination, and enables detailed diagnostics. The evaluation combines functional correctness, similarity-based metrics, and LLM-judge assessments focused on usefulness and contextual relevance. 9 state-of-the-art models were assessed, revealing differences in syntactic precision, semantic reasoning, and practical utility. Our benchmark provides actionable insights to guide model selection and improvement—detail that is often missing from other benchmarks but is essential for both practical deployment and targeted model development.

## 1 Introduction

Large Language Models (LLMs) have transformed modern software development by enabling advanced code generation, powering tools like GitHub Copilot (GitHub, 2025) and Cursor (AnySphere, 2025). As these systems are increasingly integrated into real-world workflows, realistic and rigorous evaluation frameworks are essential to understanding their strengths and limitations.

Existing benchmarks evaluate different aspects of code generation: problem solving benchmarks for coding problems (Chen et al., 2021; Austin et al., 2021; Hendrycks et al., 2021; Iyer et al., 2018; Yin et al., 2018), repository-based benchmarks for challenges in large projects (Wu et al., 2024; Du et al., 2023; Ding et al., 2023; Yu et al., 2024; Zhuo et al., 2025; Jimenez et al., 2024; Deng et al., 2025), and evolving benchmarks addressing contamination (Li et al., 2024; Jain et al., 2024).

However, existing benchmarks rely on code samples scraped from open source repositories or coding challenge websites and generate target completions based on static rules for filling in line, function, or class implementations. This limits them in several ways: First, the target completions are not based on real world usage patterns for code completion tools, and therefore do not focus on common challenging completion scenarios that arise in real world usage. Second, the diagnostic value of these benchmarks is limited because they report aggregate metrics, but cannot attribute differences in performance to specific usage areas. Third, benchmarks collected from publicly available sources are prone to training data contamination, which has been observed in models overfitting to existing benchmarks (Jain et al., 2024).

To address these limitations, we introduce **DevBench**, a realistic and scalable benchmark grounded in observed developer behavior. **DevBench** focuses on common yet challenging completion scenarios, identified from internal telemetry of over **one billion developer code completion interactions** and synthesized into 1,800 evaluation instances spanning six languages and six task categories. Each instance is reviewed for quality and realism, ensuring that tasks reflect how developers actually use code completion tools while remaining contamination-resistant.

As shown in Table 1, **DevBench** advances beyond existing benchmarks in both realism (Paul et al., 2024) and scope. It offers four key advantages: **(1) realism**, with tasks rooted in observed developer behavior; **(2) contamination resistance**, through synthetic but controlled instance generation; **(3) fine-grained evaluation**, assessing semantic alignment and developer utility; and **(4) cross-language coverage**, spanning Python (Py), JavaScript (JS), TypeScript (TS), Java, C++, and C#.

Table 1: Comparison of a collection of recent code generation benchmarks across size, language coverage, focus, source, and unique features.

| Benchmark | # Tasks | Languages | Focus | Source | Unique Feature |
|---|---|---|---|---|---|
| RepoMasterEval | 288 | Py, TS | Real-world repository completion | GitHub repos (>100 stars) | Mutation testing for test robustness |
| CrossCodeEval | ∼10k | Py, Java, TS, C# | Cross-file dependencies | GitHub repos (>3 stars) | Static analysis for dependencies |
| CoderEval | 460 | Py, Java | Cross-file pragmatic generation | GitHub repos (popular tags) | Human-labeled docstrings |
| ClassEval | 100 | Py | Class-level generation | Manually crafted | Multiple interdependent methods |
| HumanEval | 164 | Py | Basic programming tasks | Manually crafted | Simple interview-style problems |
| HumanEval+ | 164 | Py | Enhanced testing rigor | Manually crafted | $80\times$ more evaluation instances |
| LiveCodeBench | 511 | Py | Contamination-free evaluation | Competition platforms | Time-based contamination tracking |
| SWE-bench | 2,294 | Py | Repository-level bug fixing | GitHub issues and PRs | Real-world issues from 12 popular repos |
| BigCodeBench | 1,140 | Py | Diverse function calls as tools | Human-LLM collaborative generation | 723 function calls from 139 libraries across 7 domains |
| **DevBench (this work)** | **1,800** | **Py, JS, TS, Java, C++, C#** | **Realistic developer-informed scenarios** | **Synthetically generated, manually reviewed** | **Telemetry-guided, human-validated** |

Together, these features provide ecological validity: **DevBench** reflects **authentic developer challenges** rather than hypothetical tasks, is validated through expert review, and captures diverse contexts across languages and developer skill levels. By enabling both overall rankings and scenario-specific diagnostics, **DevBench** supports informed model selection and optimization, and provides a contamination-resilient foundation for future research. We open-source the complete 1,800-instance benchmark and generation methodology.

## 2 BENCHMARK DESIGN

We view code generation as a composite, puzzle-solving task in which models must combine distinct capabilities, such as API usage, intent understanding, and syntax control. To evaluate these skills, we define benchmark categories that isolate each capability while ensuring every instance is solvable from the provided prefix/suffix, making evaluation both realistic and fair. Although individual instances are synthesized, **DevBench** is telemetry-driven: its categories, task types, and scenarios are derived from analysis of over one billion real developer interactions, with synthesis used only to instantiate these empirically derived patterns in a privacy-preserving, contamination-resistant manner.

### 2.1 FROM USER TELEMETRY TO CATEGORIES

The benchmark categories are derived from an internal telemetry dataset containing over one billion anonymized code completions, each recording the prefix, suffix, generated and golden completions, and user interactions (accept, reject, edit). This dataset spans diverse contexts over IDEs, geographical locations, language distribution, and developers ranging from students to senior engineers.

To satisfy privacy and compliance requirements, we avoid using raw user code. Instead, we construct synthetic evaluation instances that reproduce the structural complexity and usage patterns observed in telemetry. As shown in Figure 1, the benchmark-generation pipeline begins by sampling telemetry completions and annotating them to identify common failure modes, bottlenecks, and characteristic structures, which we use to derive the benchmark categories. We then refine these findings through iterative discussions with a research group that includes language specialists, ensuring the categories reflect both statistical prevalence and realistic, high-impact developer workflows.

Finally, we verify that the reviewed samples are representative, so the resulting benchmark categories capture common scenarios with clear evaluation criteria, incorporate edge cases, and present realistic challenge levels directly grounded in developer behavior.

Table 2: Language-specific adaptations. Here, ML = Machine Learning, HOFs = Higher-Order Functions, RAII = Resource Acquisition Is Initialization, and HW accel. = Hardware Acceleration.

| Category | Python | C# | C++ | Java | JavaScript | TypeScript |
|---|---|---|---|---|---|---|
| API Usage | ML libs, scientific computing | .NET Core, ASP.NET | Systems prog., graphics, HW accel. | JDK, enterprise frameworks | Browser APIs, Node.js modules | Same as JS w/ types |
| Code Purpose | Iterators/generators, context mgrs | LINQ/collections, async-await | Iterators/algorithms, multithreading | Streams/collections, lambdas | Closures, promises/async | Same as JS w/ type systems |
| Code2NL | Docstrings | XML docs | Doxygen comments | Javadoc | JSDoc | TSDoc w/ type annot. |
| Low Context | Decorators, context mgrs | Complex generics, LINQ | Template metaprog., RAII | Lambda expr., streams | Async patterns, HOFs | Adv. type features |
| Pattern Matching | Decorators, context mgrs | Generic prog., memory mgmt | Template metaprog., algorithms | Streams, HOFs | Async, event handling | Type defs, generics |
| Syntax Completion | Decorator stacking, nested contexts | LINQ expr., generic constraints | Template metaprog., RAII patterns | Stream ops, try-with-resources | Promise chaining, generators | Interface defs, adv. types |

## 2.2 BENCHMARK CATEGORIES

We define six benchmark categories based on our analysis of user telemetry. Each category targets a distinct type of developer intent and is consistently evaluated across languages, with adaptations to reflect the idioms and ecosystems of each target language (see Table 2). The categories are described in detail below (examples in Appendix A).

**API Usage**: This category tests a model's ability to correctly apply specialized library functions. Each evaluation instance consists of a prefix that sets the context, a golden completion illustrating proper API usage, and a suffix for continuation.

**Code Purpose Understanding**: This category evaluates whether a model can generate code that aligns with the underlying business logic and domain-specific conventions—not just syntactic correctness. For instance, consider a `BankAccount` class where a `withdraw` method is already implemented. We prompt the generative model to implement a new `transfer` method. Based on the existing code, the model is expected to infer the intended functionality of the new method, reuse the existing `withdraw` logic for consistency, and ensure that the amount is positive and sufficient funds are available. This task goes beyond syntactic correctness, evaluating the model's ability to reason about object-oriented design and domain-specific financial logic.

**Code2NL/NL2Code**: This category evaluates a model's ability to translate between code and natural language (NL) in both directions. This reflects real-world developer workflows, where boundaries between code and language are increasingly blurred. To align with practical use cases, our benchmark covers a wide spectrum of scenarios including: (1) NL only in the prefix, (2) code only in the prefix, (3) mixed NL and code in the prefix, (4) various NL forms including docstrings, inline comments, block comments, and user-facing documentation, and (5) different documentation styles across programming languages (e.g., Python docstrings, JavaDoc, JSDoc, XML docs, and Doxygen).

**Low Context**: This category evaluates a model's ability to complete code using minimal context (10–20 lines total), requiring it to recognize language-specific patterns and idioms. These tasks are carefully designed to be solvable despite limited information, testing the model's deep understanding of programming conventions without relying on broader context.

**Pattern Matching**: This category tests a model's ability to recognize and extend established code patterns within realistic contexts. Each test includes 2–3 clear examples in the prefix, combining a technical pattern (e.g., error handling) with a domain context (e.g., security), ensuring the model must follow the intended structure rather than generating arbitrary code.

**Syntax Completion**: This category tests a model's ability to generate complex, nested structures while adhering to language-specific syntax rules. Evaluation instances span four categories: nested control structures, complex features, multi-line patterns, and error handling. The model must correctly manage indentation, close code blocks, and match braces or parentheses, demonstrating mastery of each language's unique syntactic constructs.

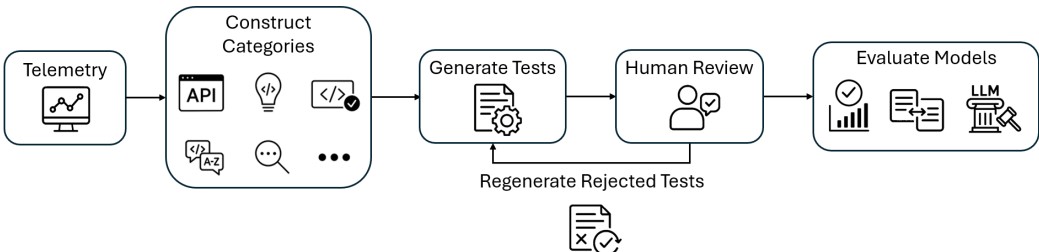

Figure 1: This diagram presents the end-to-end **DevBench** pipeline, starting with developer telemetry analysis to define six code completion categories. Evaluation instances are synthetically generated and refined through human review. Final evaluation combines functional correctness, similarity-based metrics, and LLM-based judgment to assess functional, semantic, and holistic model performance.

## 2.3 BENCHMARK CONSTRUCTION

Figure 1 overviews the construction pipeline: starting from telemetry-derived categories, we generate the benchmark.

**Evaluation Instance Structure**: Each instance consists of four components: (1) a **prefix** providing the preceding code context, (2) a **golden completion** as the expected model output, (3) a **suffix** representing subsequent code, and (4) **assertions** to validate correctness. For Java, C#, and C++, assertions are embedded in the suffix, while for Python, JavaScript, and TypeScript, they are placed in a separate section.

**Completion Modes**: **DevBench** covers key scenarios observed in real developer interactions, including both prefix-only completions and fill-in-the-middle (FIM) cases where a suffix is provided. Completions are positioned at natural code boundaries (e.g., after operators, function calls, variable declarations) reflecting realistic developer cursor positions.

**Generation and Validation**: Synthetic instances were generated with OpenAI's GPT-4o, chosen for its fluency, reasoning, and code generation capabilities (OpenAI et al., 2024). Generation used temperature 0.7 with a 4000-token limit. Each synthetic instance was first screened via automatic syntax checks, validated for functional correctness by executing the combined prefix, golden completion, suffix, and assertions to ensure all assertions pass successfully, then manually reviewed using a custom annotation tool.

**Human Review**: Each instance was independently reviewed by two annotators from a team of three senior researchers and engineers with expertise across all six target languages. The annotators evaluated four dimensions: (1) *usefulness* (if the completion satisfies a plausible developer need), (2) *realism* (if it reflects authentic coding patterns, including common inconsistencies and suboptimal but valid approaches), (3) *category alignment* (if it is consistent with the intended task type), and (4) *complexity authenticity* (if it captures the genuine difficulty and edge cases observed in telemetry). Disagreements were resolved through discussion with the third annotator, achieving reliable consensus across evaluations.

**Iteration**: Annotators were specifically instructed to prioritize realism over idealized implementations. For example, API Usage cases were validated not only for correct library calls but also for realistic parameter handling, error conditions, and incomplete context that developers actually encounter in practice. The rejected samples, primarily due to low challenge or category mismatch, were regenerated and re-verified until they met realism standards. Common rejection reasons included: overly simplified or "textbook-perfect" implementations (32% of rejections), insufficient complexity relative to telemetry-observed patterns (28%), unrealistic examples that ignored common edge cases or error conditions (23%), and category misalignment where the completion didn't match the intended task type (17%).

**Complexity**: To assess the complexity of DevBench, we report *lines of code (LOC)*, *token counts*, and *cyclomatic complexity* (Landman et al., 2016). As shown in Table 3 and Table 4, DevBench offers higher complexity and realism than prior benchmarks, with evaluation instances averaging

Table 3: Complexity of code generation benchmarks.

| Metric | DevBench (ours) | CrossCodeEval | CoderEval-Py | CoderEval-Java | APPS | HumanEval | MBPP | Concode | CoNaLA | DS-1000 |
|---|---|---|---|---|---|---|---|---|---|---|
| **Avg. LOC** | 65.3 | 71.1–116.5 | 32.0 | 10.2 | 21.4 | 11.5 | 6.8 | 4.8 | 1.0 | 3.8 |
| **Cyclomatic Complexity** | 5.5 | – | 4.7 | 3.1 | – | 3.6 | – | 1.4 | – | – |

Table 4: DevBench language-specific statistics.

| Language | Prefix LOC | Completion LOC | Total LOC | Prefix Tokens | Completion Tokens | Cyclomatic |
|---|---|---|---|---|---|---|
| Python | 19.5 | 4.2 | 40.5 | 92.6 | 21.8 | 2.2 |
| C# | 49.9 | 5.9 | 73.4 | 190.3 | 35.7 | 5.0 |
| C++ | 43.8 | 5.4 | 63.0 | 199.0 | 41.8 | 6.2 |
| Java | 34.9 | 5.1 | 55.5 | 154.5 | 37.2 | 4.6 |
| JavaScript | 41.7 | 6.7 | 70.8 | 227.5 | 48.3 | 6.7 |
| TypeScript | 58.7 | 10.7 | 88.4 | 319.1 | 80.7 | 8.2 |
| **Average** | **41.4** | **6.3** | **65.3** | **197.2** | **44.2** | **5.5** |

65.3 LOC and a cyclomatic complexity of 5.5. Importantly, DevBench maintains a balanced prefix-to-completion ratio: completions average 6.3 LOC, with 197.2 tokens in the prefix and 44.2 in the completion. In contrast, CrossCodeEval features long prompts (71–116 LOC) but extremely short completions (1–2 LOC). This balance makes DevBench more reflective of practical code-completion workflows, where both context and generated code contribute meaningfully to task complexity.

**Bias Mitigation**: To address potential systematic bias from using GPT-4o as the generator, we note that recent studies suggest GPT-4o introduces minimal stylistic bias (Maheshwari et al., 2024; Chen et al., 2024), and our human validation process further mitigates risks. Empirically, our evaluation results demonstrate that the benchmark does not favor GPT-family models: multiple non-GPT models (e.g., Claude 4 Sonnet, Claude 3.7 Sonnet) outperform GPT-4o on **DevBench** (Section 4), indicating that generator bias is minimal and does not affect evaluation fairness.

## 3 EVALUATION METHODS

Given the challenges in evaluating LLMs, we employ a combination of methods: functional correctness; similarity-based metrics, which offer fast, scalable evaluation across languages; and LLM-judge evaluations to assess output quality from a human-aligned perspective.

### 3.1 FUNCTIONAL CORRECTNESS

For functional correctness, we report Pass@1 with $n = 5$ samples (Chen et al., 2021), measuring the probability that at least one generated sample passes all test cases:

$$\text{pass@}k := \mathbb{E}_{\text{Problems}} \left[ 1 - \frac{\binom{n-c}{k}}{\binom{n}{k}} \right] \tag{1}$$

where $c$ is the number of correct samples and $k = 1$ (Execution details are in Appendix E.2).

### 3.2 SIMILARITY-BASED EVALUATION

We use two widely adopted similarity metrics: Average Cosine Similarity and Line 0 Exact Match Rate. Average Cosine Similarity assesses semantic equivalence across the full completion, even when syntax differs, while Line 0 Exact Match focuses on strict precision at the start of the completion. Each metric is averaged over the $n = 5$ generated completions per test case.

**Average Cosine Similarity**: We use token-based cosine similarity (Zhou et al., 2023) to measure semantic overlap between model-generated and golden completions. When tokenization fails due to unusual code constructs, we fall back to character n-grams (1-3) to ensure robust comparison.

**Line 0 Exact Match Rate**: We calculate the percentage of cases where the first line of the model-generated completion exactly matches the first line of the golden completion (Ding et al., 2023)

### 3.3 LLM-JUDGE EVALUATION

For automated code evaluation, we designed an LLM-based judge that scores each completion along two dimensions: **relevance** to the provided context and **helpfulness** in advancing the task. Each aspect is rated on a 0–5 scale, yielding a combined score from 0 to 10. For each test case, we generate $n = 5$ completions and average the judge scores to obtain a single quality score per test case.

**Bias Mitigation**: We use o3-mini (OpenAI et al., 2025c) as the LLM judge, chosen for its favorable bias profile: according to the OpenAI system card, it exhibits the lowest measured bias among comparable models on discrimination tasks (Tong & Zhang, 2024). To further limit evaluation bias, we blind the judge to the generating model's identity and architecture. Moreover, we align the judge with human judgments (see below), providing an additional layer of bias mitigation.

**Human-LLM Agreement Validation**: We iteratively tuned the prompt on telemetry acceptance signals from 10,000 completions (accepted as-is, rejected, edited after acceptance), selecting relevance and helpfulness criteria that maximized Spearman correlation. We then validated on a stratified set of 150 completions (25 per language) with three experienced annotators scoring on a 0–10 rubric, averaging after acceptable inter-annotator agreement. Human ratings correlated strongly with o3-mini judgments, indicating alignment with developer preferences at scale.

**Confidence Intervals**: For each model, we compute average scores by programming language and evaluation scenario. We then aggregate completions within each language to obtain an overall average and a 95% confidence interval, estimated via 10,000 bootstrap resamples. Finally, we report the overall average score across all languages and completions with its corresponding confidence interval.

## 4 EXPERIMENTS

### 4.1 EXPERIMENTAL SETUP

**Models**: We evaluated a diverse set of state-of-the-art LLMs to capture code generation performance across varying training approaches and scales. Our selection included multiple OpenAI models (OpenAI et al., 2025a;b), Anthropic's Claude Sonnet series (Anthropic, 2025; Cla), DeepSeek models (DeepSeek-AI et al., 2025), and Ministral 3B (Mistral, 2025) as a representative compact open model.

**Evaluation Setup**: Following prior work (Jain et al., 2024), we used a temperature of 0.2 for more deterministic completions and set a maximum output length of 800 tokens to accommodate complex completions. All models used nucleus sampling with top-p=1.0 to preserve the full token distribution while modulating randomness via temperature. For LLM-judge evaluation, we used o3-mini as a strong reasoning model with default settings (temperature=1.0 and top-p=1.0). Models were evaluated in a zero-shot setting, each prompted using a consistent, code-only template, excluding explanations or comments. See Appendix E.4 for prompt details.

Infrastructure details including hardware specifications and execution environments are provided in Appendix E.1.

### 4.2 RESULTS AND INSIGHTS

#### 4.2.1 FUNCTIONAL CORRECTNESS (PASS@1)

Table 5 shows Pass@1 results with $n = 5$ samples, averaged across the six programming languages. Language-specific Pass@1 breakdowns are provided in Table 9.

**Top Performers**: Claude 4 Sonnet leads with 84.80%, followed by Claude 3.7 Sonnet (80.60%) and GPT-4.1 mini (79.70%).

Table 5: Pass@1 with $n = 5$ across code completion categories.

| Model | Overall ↓ | API Usage | Code Purpose | Code2NL/NL2Code | Low Context | Pattern Matching | Syntax |
|---|---|---|---|---|---|---|---|
| Claude 4 Sonnet | 84.80% | 87.50% | 86.50% | 78.90% | 90.30% | 81.00% | 84.70% |
| Claude 3.7 Sonnet | 80.60% | 81.90% | 84.90% | 68.90% | 89.80% | 75.70% | 82.00% |
| GPT-4.1 mini | 79.70% | 82.70% | 84.10% | 66.00% | 87.10% | 76.70% | 81.60% |
| GPT-4.1 | 78.70% | 84.30% | 84.70% | 60.80% | 88.20% | 73.80% | 80.20% |
| GPT-4o | 77.20% | 82.30% | 83.60% | 58.20% | 87.50% | 71.10% | 80.30% |
| DeepSeek-V3 | 75.30% | 78.10% | 83.00% | 57.50% | 82.70% | 73.30% | 77.30% |
| DeepSeek-V3.1 | 72.00% | 77.50% | 79.10% | 54.60% | 81.30% | 68.60% | 71.00% |
| GPT-4.1 nano | 66.40% | 69.80% | 73.10% | 48.50% | 72.60% | 62.70% | 71.80% |
| Ministral 3B | 48.60% | 53.10% | 57.80% | 33.50% | 51.00% | 44.10% | 52.30% |

**Small-Size Models**: Ministral-3B achieves a modest overall Pass@1 of 48.60%, while GPT-4.1 nano reaches 66.40%.

**Category Performance Patterns**: *Low Context* emerges as the strongest category across models, with top performers achieving 87-90% success rates. Conversely, *Code2NL/NL2Code* represents the most challenging category, where even leading models like Claude 4 Sonnet achieve 78.90%, and most others fall below 70%.

**Pattern Matching**: This category shows significant model differentiation, with Claude 4 Sonnet achieving 81.00% while smaller models like Ministral-3B and GPT-4.1 nano lag at 44.10% and 62.70% respectively. This demonstrates that pattern recognition and extension capabilities serve as key differentiators between model tiers.

**API Usage**: Strong performance across top-tier models, with Claude 4 Sonnet (87.50%), GPT-4.1 (84.30%), and GPT-4.1 mini (82.70%) leading. However, the 35+ percentage point gap between top performers and Ministral-3B (53.10%) highlights the complexity of correctly applying specialized library functions.

### 4.2.2 SIMILARITY-BASED EVALUATION

Table 6 reports similarity metrics across languages, averaged over categories. Additional similarity results are available in Appendix E.5.

**Top Performers**: DeepSeek-V3 demonstrates strong performance across both similarity metrics, achieving some of the highest Average Cosine Similarity scores (0.70 in Python, 0.75 in Java). GPT-4.1 mini also shows consistent performance with balanced scores across languages. Claude 4 Sonnet exhibits competitive performance, particularly excelling in Java with 65.0% Line 0 Exact Match Rate.

**Metric Discrepancies**: In Pattern Matching, Claude 3.7 Sonnet achieves 75.70% on Pass@1 but shows more modest similarity scores, while DeepSeek-V3 demonstrates higher Average Cosine Similarity across languages (0.70 in Python vs 0.64, 0.75 in Java vs 0.71) and stronger Line 0 Exact Match Rates (50.33% in Python vs 46.33%, 64.0% in Java vs 60.33%) despite lower Pass@1 performance (73.30%). This discrepancy helps identify areas where deeper analysis is needed to understand model behavior. For example, DeepSeek-V3 reliably replicates familiar code patterns but sometimes fails to maintain full functional correctness, while Claude's solutions, while functionally correct, frequently employ alternative implementation approaches that diverge syntactically from the reference solutions which match patterns.

**Language-Specific Challenges**: TypeScript consistently emerges as the most challenging language, with most models showing 20-30% lower performance compared to other languages. This consistent difficulty stems from its complex type system and the need to maintain strict type consistency throughout the code.

Please see Appendix B and Appendix D for qualitative examples of the mentioned behaviors.

### 4.2.3 LLM-JUDGE EVALUATION

Figure 2 presents the final LLM-judge scores with 95% confidence intervals and Appendix C provides a detailed breakdown by category and language.

Table 6: Similarity metrics across programming languages.

| Model | Average Cosine Similarity | | | | | | Line 0 Exact Match Rate (%) | | | | | |
|---|---|---|---|---|---|---|---|---|---|---|---|---|
| | Py | JS | TS | Java | C++ | C# | Py | JS | TS | Java | C++ | C# |
| Claude 4 Sonnet | 0.68 | 0.56 | 0.53 | 0.74 | 0.72 | 0.65 | 51.0 | 44.33 | 38.67 | 65.0 | 59.0 | 50.67 |
| Claude 3.7 Sonnet | 0.64 | 0.57 | 0.50 | 0.71 | 0.71 | 0.64 | 46.33 | 45.0 | 36.33 | 60.33 | 58.33 | 50.33 |
| GPT-4.1 mini | 0.71 | 0.57 | 0.50 | 0.72 | 0.72 | 0.65 | 53.0 | 45.33 | 36.33 | 57.0 | 60.0 | 48.0 |
| GPT-4.1 | 0.68 | 0.57 | 0.50 | 0.71 | 0.72 | 0.65 | 48.67 | 45.0 | 33.33 | 59.67 | 60.33 | 48.67 |
| GPT-4o | 0.68 | 0.58 | 0.53 | 0.72 | 0.72 | 0.64 | 48.0 | 46.0 | 40.0 | 60.67 | 60.0 | 46.33 |
| DeepSeek-V3 | 0.70 | 0.58 | 0.57 | 0.75 | 0.73 | 0.66 | 50.33 | 44.67 | 43.33 | 64.0 | 59.0 | 48.67 |
| DeepSeek-V3.1 | 0.71 | 0.54 | 0.51 | 0.70 | 0.72 | 0.63 | 53.67 | 46.0 | 41.33 | 61.0 | 62.33 | 48.67 |
| GPT-4.1 nano | 0.59 | 0.49 | 0.42 | 0.64 | 0.64 | 0.55 | 38.33 | 32.67 | 28.67 | 46.33 | 49.33 | 35.67 |
| Ministral 3B | 0.51 | 0.40 | 0.35 | 0.54 | 0.52 | 0.49 | 28.67 | 22.67 | 18.0 | 35.0 | 35.0 | 30.67 |

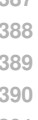

Figure 2: Overall LLM-judge evaluation scores with 95% confidence intervals.

**High Performers**: GPT-4o leads with the highest LLM-judge score, followed closely by DeepSeek-V3 and Claude 3.7 Sonnet, revealing a ranking that differs from other evaluation methods.

**Reasoning vs Non-Reasoning Models**: The results show a mixed pattern between reasoning and non-reasoning capabilities. While models with reasoning capabilities, like DeepSeek-V3 and Claude 3.7 Sonnet, rank highly, the top performer is GPT-4o, a non-reasoning model. Interestingly, Claude 4 Sonnet, despite having reasoning capabilities with the highest Pass@1 scores (84.80%), ranks lower in LLM-judge evaluation, suggesting that reasoning capabilities may enhance functional correctness but don't necessarily align with the judge's criteria for code relevance and helpfulness.

**Confidence Intervals**: Most models display relatively narrow confidence intervals, indicating consistent performance across evaluation instances. However, Claude 4 Sonnet shows wider confidence intervals, indicating greater variability and reduced consistency—particularly on complex or edge-case prompts—even when its average performance is competitive.

## 4.3 DIAGNOSTIC CASE STUDY: DEEPSEEK-V3

**DevBench**'s multi-metric framework enables fine-grained diagnosis beyond aggregate rankings. To demonstrate its practical utility, we present a case study on DeepSeek-V3: although its overall performance is competitive, our analysis identifies specific opportunities for improvement.

**Syntax vs. Semantics:** DeepSeek-V3 excels in Pattern Matching similarity in Table 8 (Average Cosine Similarity 0.75 vs. Claude 3.7 Sonnet's 0.70 and Line 0 Exact Match Rate 60.0% vs. 53.0%) but underperforms in functional correctness of the same category in Table 5 (73.30% vs. 75.70% Pass@1). This pattern indicates heavier reliance on pattern memorization than true semantic understanding. Manual review of failure cases confirms that DeepSeek-V3 often produces code syntactically close to the golden solution but functionally incorrect.

**Category-Level Patterns:** Based on Table 8, the model demonstrates strong performance in Pattern Matching (0.75 vs 0.70) and Syntax Completion (0.65 vs. 0.59) but underperforms in Code2NL/NL2Code tasks (0.53 vs. 0.59). This disparity reveals the model's tendency to memorize surface patterns rather than deeply understand and generate code in semantically rich tasks requiring bidirectional translation between natural language and code.

**Language-Specific Gaps:** While DeepSeek-V3 performs competitively in Python (72.7%) and Java (85.7%), it shows notable underperformance in C++ (77.8%, ranking 7th) (Table 9). Our cross-model analysis suggests targeted improvements in C++ could yield broader gains.

**Preserving Strengths:** DeepSeek-V3 already excels in Syntax Completion and Python development, areas that should be maintained during future fine-tuning to avoid catastrophic forgetting.

These insights translate to actionable training priorities: (1) emphasize pattern extension and reasoning during fine-tuning to reduce over-reliance on memorization, (2) increase Code2NL/NL2Code training examples to improve semantic understanding, (3) include more C++ samples in the training mix to close performance gaps, and (4) maintain current strengths in Python and Syntax Completion.

## 5 RELATED WORK

Existing LLM coding evaluation spans three main areas. *Problem solving benchmarks* like HumanEval (Chen et al., 2021), MBPP (Austin et al., 2021), and APPS (Hendrycks et al., 2021) evaluate coding problems of varying difficulty, while Concode and CoNaLa focus on natural language to code translation (Iyer et al., 2018; Yin et al., 2018). *Repository-based benchmarks* evaluate code generation within existing codebases, from simple masking tasks (RepoMasterEval, ClassEval (Wu et al., 2024; Du et al., 2023)) to inter-file reasoning (CrossCodeEval, CoderEval (Ding et al., 2023; Yu et al., 2024)) and API usage (BigCodeBench (Zhuo et al., 2025)). SWE-Bench extends this to agentic problem solving (Jimenez et al., 2024), and SWE-Bench Pro (Deng et al., 2025) introduces more challenging enterprise-level problems with contamination resistance through GPL licensing and commercial codebases. *Evolving benchmarks* like LiveCodeBench and EvoCodeBench address data contamination using recent code (Jain et al., 2024; Li et al., 2024).

In contrast, **DevBench** evaluates scenarios arising during live development rather than repository-based or challenge problems.

## 6 CONCLUSION

We introduced **DevBench**, a synthetic benchmark grounded in developer telemetry, enabling fine-grained, realistic code completion evaluation across six languages and task categories, resulting in 1,800 evaluation instances with a focus on ecological validity, contamination resistance, and interpretability. Evaluating 9 state-of-the-art models, we observed consistent strengths in low-context pattern recognition and persistent challenges in bidirectional natural language–code translation and syntactic alignment. Our multi-pronged evaluation—combining functional correctness, similarity metrics, and LLM-judge assessments—revealed nuanced differences such as cross-language consistency and robustness across task types. By releasing the benchmark and its generation infrastructure, we aim to support the research community in advancing more accountable, targeted, and practical evaluation of code generation models.

Future work could explore developing composite evaluation metrics that capture the full spectrum of code quality dimensions and broadening coverage scope by applying our methodology to additional development activities such as code refactoring, debugging, and multi-file architecture design.

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

## A  FURTHER CATEGORY EXAMPLES

### A.1  API USAGE

To illustrate this category, consider the following Python example that evaluates a model's ability to correctly implement asynchronous HTTP requests using the Tornado library:

**Example 1: Python API Usage #1**

```python
import asyncio
from tornado.httpclient import AsyncHTTPClient

async def fetch_url(url):
    client = AsyncHTTPClient()
    try:
```

**Prefix**

```python
        response = await client.fetch(url)
```

**Golden Completion**

```
    finally:
        client.close()
    return response.body
    ...
```

**Suffix**

This example demonstrates several key aspects of our API Usage category. First, it tests knowledge of proper asynchronous programming in Python using Tornado's HTTP client. The model must correctly implement the `await` keyword with the appropriate API method (`client.fetch`). The prefix establishes the asynchronous context and resource acquisition, while the suffix ensures proper resource cleanup via the `finally` block. This pattern of establishing context, implementing specialized API calls, and ensuring proper resource management is representative of real-world coding scenarios that developers encounter when working with external libraries.

## A.2 CODE2NL/NL2CODE

The following example demonstrates the Code2NL aspect of this category, where a model must generate appropriate documentation for a C++ financial class:

### Example 2: C++ Code2NL/NL2Code #4

```
#include <iostream>
#include <string>
#include <vector>
#include <cassert>
```

**Prefix**

```
/**
 * @brief Represents a financial instrument in a portfolio.
 * @param name The name of the instrument.
 * @param price The price of the instrument.
 * @param risk The risk factor of the instrument.
 */
```

**Golden Completion**

```
class FinancialInstrument {
public:
    FinancialInstrument(std::string name, double price, double risk) :
    name_(name), price_(price), risk_(risk) {}

    /// @brief Gets the name of the instrument.
    /// @return The name of the instrument.
    std::string getName() const { return name_; }

    /// @brief Gets the price of the instrument.
    /// @return The price of the instrument.
    double getPrice() const { return price_; }

    /// @brief Gets the risk factor of the instrument.
    /// @return The risk factor of the instrument.
    double getRisk() const { return risk_; }

private:
    std::string name_;
    double price_;
    double risk_;
};

...
```

**Suffix**

In this example, the model must generate Doxygen-style documentation for a C++ class constructor by inferring the class's purpose and parameters from the subsequent implementation. The prefix contains only standard C++ includes, providing minimal context, while the suffix shows the full class implementation with member functions already documented. The model must understand both the domain context (financial instruments in a portfolio) and the Doxygen documentation format, including the appropriate use of `@brief` for the class description and `@param` tags for each constructor parameter. This tests the model's ability to extract semantic meaning from implementation details and express it in standardized documentation format. The model must "reverse engineer" code understanding into NL explanation, following language-specific documentation conventions that would be expected in professional C++ codebases.

### A.3 LOW CONTEXT

The following example demonstrates how this category tests programming pattern recognition with minimal context in C#:

**Example 3: C# Low Context #47**

```csharp
using System;
using System.Collections.Generic;
using System.Diagnostics;

public class CustomPaginator<T>
{
    private readonly List<T> _items;
    private readonly int _pageSize;

    public CustomPaginator(List<T> items, int pageSize)
    {
        _items = items;
        _pageSize = pageSize;
    }

    public IEnumerable<IEnumerable<T>> GetPages()
    {
```

**Prefix**

```csharp
        for (int i = 0; i < _items.Count; i += _pageSize)
        {
            yield return _items.GetRange(i, Math.Min(_pageSize, _items.
Count - i));
```

**Golden Completion**

```csharp
        }
    }
}

...
```

**Suffix**

In this example, the model must implement a pagination iterator with minimal surrounding context. With just the class structure, field definitions, and method signature, the model must infer that an iterator pattern using C#'s `yield return` statement is the idiomatic approach for implementing a paginator. The golden completion demonstrates key C# idioms: using a `for` loop for pagination control, calculating page boundaries with `Math.Min()` to handle the last page case, and most

importantly, using the `yield return` pattern to create a lazy enumeration of pages. This pattern is specific to C# and allows for deferred execution of the pagination logic. The model must recognize from the return type `IEnumerable<IEnumerable<T>>` that the method should return a sequence of sequences without materializing the entire result set at once. This example highlights how even with minimal context (less than 15 lines total), models must demonstrate deep understanding of language-specific patterns and implement idiomatic solutions that align with the established programming conventions for each language.

A.4   SYNTAX COMPLETION

The following example demonstrates how this category tests understanding of Java's modern functional syntax features:

**Example 4: Java Syntax Completion #5**

```java
import java.util.Optional;

public class User {
    private String name;
    private String email;

    public User(String name, String email) {
        this.name = name;
        this.email = email;
    }

    public String getName() {
        return name;
    }

    ...
}

public class UserService {
    ...

    public String getUserNameOrDefault(String email) {
        Optional<User> userOpt = findUserByEmail(email);
```

**Prefix**

```java
        return userOpt.map(User::getName).orElse("Unknown User");
```

**Golden Completion**

```java
    }

    public static void main(String[] args) {
        UserService userService = new UserService();
        String userName = userService.getUserNameOrDefault("test@example.
com");
        assert userName.equals("Test User") : "Expected 'Test User', but
got " + userName;
        userName = userService.getUserNameOrDefault("unknown@example.com"
);
        assert userName.equals("Unknown User") : "Expected 'Unknown User
', but got " + userName;
    }
}
```

**Suffix**

In this example, the model must demonstrate understanding of Java's `Optional` API and method chaining syntax, which are modern Java features introduced to handle nullable values functionally. The prefix establishes a scenario where a user might be found by email address, returning an `Optional<User>` that could be empty. The golden completion showcases several Java-specific syntax elements in a single line: the functional-style `map` operation with method reference syntax (`User::getName`), followed by chained method invocation with the terminal operation `orElse` to provide a default value. This completion requires precise syntax understanding as it involves multiple Java-specific features: proper method chaining, correct use of method references, and appropriate handling of the `Optional` container. The suffix validates the implementation with assertions testing both the successful case and the default fallback. This example tests the model's ability to produce syntactically correct Java code that leverages modern language features, demonstrating mastery beyond basic language syntax. The conciseness of the golden completion—accomplishing a common nullable-handling pattern in a single expressive line—is representative of idiomatic modern Java programming that models should be capable of generating.

## A.5 PATTERN MATCHING

The following example illustrates how this category tests pattern recognition in a Java functional programming context:

**Example 5: Java Pattern Matching #29**

```java
import java.util.List;
import java.util.ArrayList;
import java.util.function.Function;

// This class demonstrates the use of higher-order functions to apply
    different transformations to a list of integers
public class HigherOrderFunctionsDemo {

    // Method to apply a transformation to a list of integers
    public static List<Integer> transformList(List<Integer> list,
    Function<Integer, Integer> transformation) {
        ...
    }

    // Sample transformations
    public static Function<Integer, Integer> square = x -> x * x;
    public static Function<Integer, Integer> cube = x -> x * x * x;
    public static Function<Integer, Integer> negate = x -> -x;

    public static void main(String[] args) {
        List<Integer> numbers = new ArrayList<>();
        for (int i = 1; i <= 5; i++) {
            numbers.add(i);
        }

        // Apply the square transformation
        List<Integer> squaredNumbers = transformList(numbers, square);
        System.out.println("Squared Numbers: " + squaredNumbers);

        // Apply the cube transformation
        List<Integer> cubedNumbers = transformList(numbers, cube);
        System.out.println("Cubed Numbers: " + cubedNumbers);
```

**Prefix**

```java
        // Apply the negate transformation
        List<Integer> negatedNumbers = transformList(numbers, negate);
        System.out.println("Negated Numbers: " + negatedNumbers);
```

**Golden Completion**

```
        // Assertions
        assert squaredNumbers.equals(List.of(1, 4, 9, 16, 25)) : "Squared
   numbers are incorrect";
        assert cubedNumbers.equals(List.of(1, 8, 27, 64, 125)) : "Cubed
   numbers are incorrect";
        assert negatedNumbers.equals(List.of(-1, -2, -3, -4, -5)) : "
   Negated numbers are incorrect";
    }
}
```

**Suffix**

In this task, the model extends a functional programming pattern in Java, where higher-order functions are applied to transform a list of integers. The prefix includes two examples (squaring and cubing) using a consistent structure: defining a transformation, applying it via `transformList`, and printing results with a descriptive message. The model must follow this structure and semantics to implement a third transformation (`negate`). This tests the model's ability to recognize and continue idiomatic Java patterns using higher-order functions in a well-defined context.

## A.6 CODE PURPOSE UNDERSTANDING

The following example illustrates and example in the financial domain context:

**Example 6: Python Code Purpose Understanding #5**

```python
class BankAccount:
    def __init__(self, account_number, balance=0):
        self.account_number = account_number
        self.balance = balance

    ...

    def withdraw(self, amount):
        if amount > 0 and amount <= self.balance:
            self.balance -= amount
            return self.balance
        else:
            raise ValueError("Insufficient funds or invalid withdrawal
   amount")

    def transfer(self, target_account, amount):
```

**Prefix**

```python
        if amount > 0 and amount <= self.balance:
            self.withdraw(amount)
        else:
            raise ValueError("Insufficient funds or invalid transfer
   amount")
```

**Golden Completion**

```python
        target_account.deposit(amount)
        return self.balance

...
```

**Suffix**

In this task, the model implements the `transfer` method in a `BankAccount` class, requiring it to validate that the amount is positive and funds are sufficient. It must reuse the existing `withdraw` method for consistency and raise a domain-specific error if validation fails. This goes beyond syntax, testing the model's ability to reason about object-oriented structure and financial business logic.

## B ILLUSTRATIVE MODEL COMPARISON

**Example 7: Python Pattern Matching #18**

```python
class DataValidator:
    def __init__(self, schema):
        self.schema = schema

    def validate(self, data):
        errors = []
        for field, rules in self.schema.items():
            if field not in data:
                errors.append(f'Missing field: {field}')
            else:
                value = data[field]
                for rule, expected in rules.items():
                    if rule == 'type' and not isinstance(value, expected)
    :
                        errors.append(f'Field {field} should be of type {
    expected.__name__}')
                    elif rule == 'min_length' and len(value) < expected:
                        errors.append(f'Field {field} should be at least
    {expected} characters long')
```

**Prefix**

```python
                    elif rule == 'max_length' and len(value) > expected:
                        errors.append(f'Field {field} should be at most {
    expected} characters long')
```

**Golden Completion**

```python
                    elif rule == 'max_length' and len(value) > expected:
                        errors.append(f'Field {field} should be at most {
    expected} characters long')
                    elif rule == 'required' and not value:
                        errors.append(f'Field {field} is required')
```

**Model Completion (DeepSeek-V3)**

```python
                    elif rule == 'max_length' and len(value) > expected:
                        errors.append(f'Field {field} should be at most {
    expected} characters long')
                    elif rule == 'min_value' and value < expected:
                        errors.append(f'Field {field} should be at least
    {expected}')
                    elif rule == 'max_value' and value > expected:
                        errors.append(f'Field {field} should be at most {
    expected}')
```

**Model Completion (Claude 3.7 Sonnet)**

```python
        return errors

schema = {
    'name': {'type': str, 'min_length': 3},
    'age': {'type': int}
}

validator = DataValidator(schema)
data_valid = {'name': 'Alice', 'age': 30}
data_invalid = {'name': 'Al', 'age': 'thirty'}

errors_valid = validator.validate(data_valid)
```

```
errors_invalid = validator.validate(data_invalid)

assert errors_valid == []
assert errors_invalid == ['Field name should be at least 3 characters
    long', 'Field age should be of type int']

print('All assertions passed!')
```

**Suffix**

```
schema = {
    'name': {'type': str, 'min_length': 3, 'max_length': 10},
    'age': {'type': int}
}

validator = DataValidator(schema)
data_valid = {'name': 'Alice', 'age': 30}
data_invalid_length = {'name': 'Aliceeeeeeee', 'age': 30}
data_invalid_type = {'name': 'Alice', 'age': 'thirty'}

errors_valid = validator.validate(data_valid)
errors_invalid_length = validator.validate(data_invalid_length)
errors_invalid_type = validator.validate(data_invalid_type)

assert errors_valid == []
assert errors_invalid_length == ['Field name should be at most 10
    characters long']
assert errors_invalid_type == ['Field age should be of type int']
```

**Assertions**

The prefix code presents a `DataValidator` class that implements a validation framework for checking data against a schema. The class has been partially implemented with methods to initialize the validator and validate data, including checks for missing fields, type validation, and minimum length validation. The established pattern is evident in the validation logic structure, where each rule check follows a consistent "if/elif" pattern with appropriate error messages. This demonstrates a real-world scenario where consistent validation rules are essential for maintaining data integrity. The golden completion adds a single rule check for `max_length` that follows the established pattern exactly, validating that field values do not exceed a maximum length and generating an appropriate error message that matches the style of previous validation checks. This completion perfectly extends the pattern established in the prefix and is required to satisfy the assertions. DeepSeek-V3's completion correctly implements the required `max_length` validation rule, matching the golden completion exactly, and then adds only one additional validation rule for the required property. While this additional check is unnecessary for passing the assertions, DeepSeek-V3's completion remains relatively close to the golden standard by limiting its extension to a single additional validation rule that follows the established pattern. Claude 3.7 Sonnet's completion also correctly implements the required `max_length` validation, but then extends the pattern with two unnecessary additional rules for `min_value` and `max_value` validation. These additional rules, while following the established pattern and potentially useful in a real validation system, represent a more significant deviation from the golden completion compared to DeepSeek-V3's response. The inclusion of these two extra validation rules makes Claude 3.7 Sonnet's completion less similar to the golden standard. The suffix code and assertions validate the functionality, confirming that the required `max_length` validation is essential for passing the tests. The example illustrates why DeepSeek-V3 demonstrates stronger Average Cosine Similarity in Pattern Matching compared to Claude 3.7 Sonnet: it more closely adheres to the minimal required pattern extension by adding fewer unnecessary validation rules.

## C    DETAILED LLM-JUDGE EXPERIMENTAL RESULTS

The heat-map in Figure 3 shows the breakdown of LLM-judge scores by category and languages.

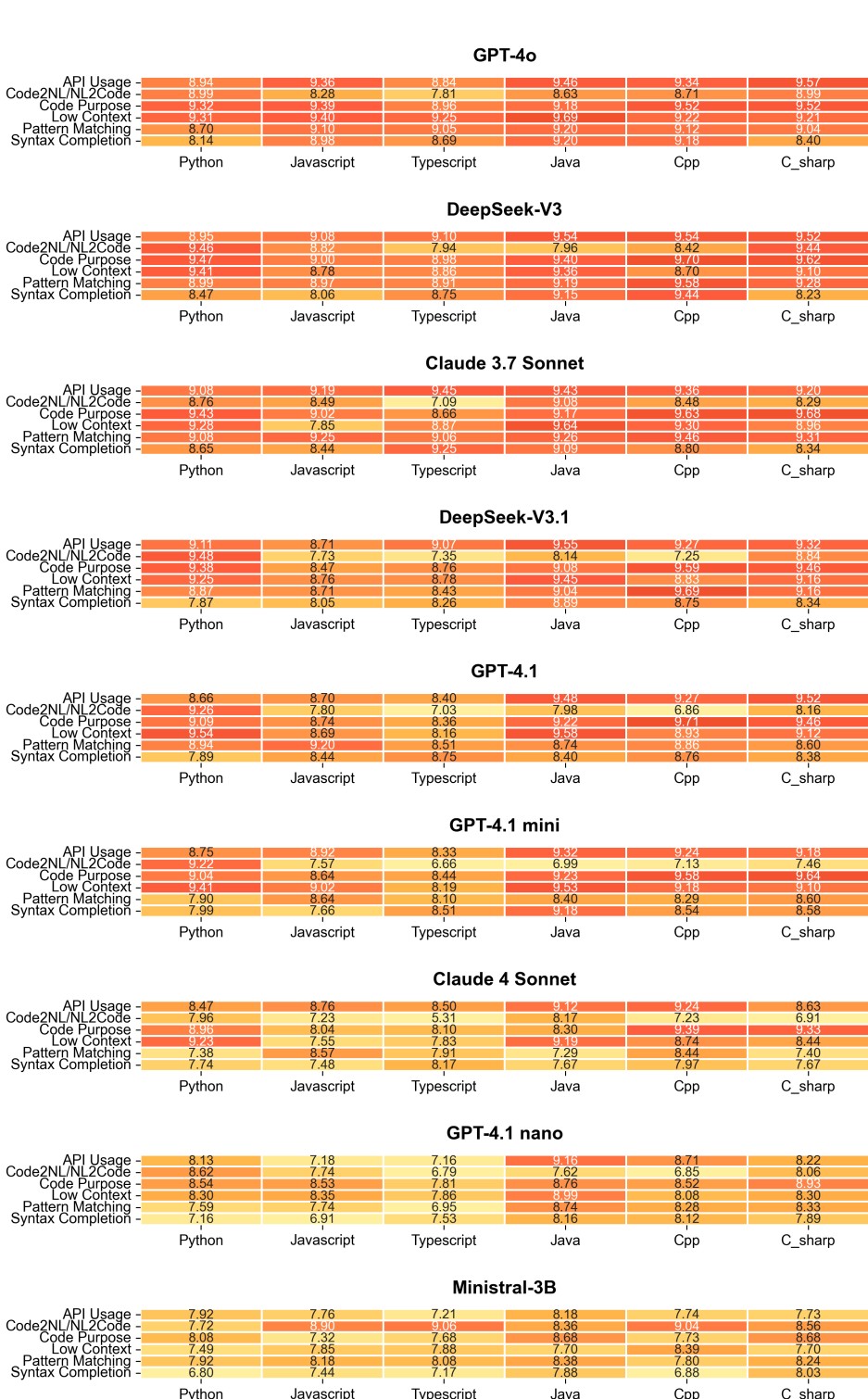

Figure 3: Breakdown of LLM-judge scores across models.

Table 7: Programming language LLM-judge scores of different LLMs with 95% confidence intervals.

| Model | C++ | C# | Java | JavaScript | Python | TypeScript |
|---|---|---|---|---|---|---|
| Claude 4 Sonnet | 8.50 (8.16-8.82) | 8.06 (7.66-8.45) | 8.29 (7.90-8.67) | 7.94 (7.55-8.32) | 8.29 (7.93-8.64) | 7.64 (7.23-8.02) |
| Claude 3.7 Sonnet | 9.17 (8.94-9.39) | 8.96 (8.69-9.21) | 9.28 (9.06-9.48) | 8.71 (8.41-8.99) | 9.05 (8.81-9.28) | 8.73 (8.43-9.00) |
| GPT-4.1 mini | 8.66 (8.36-8.95) | 8.76 (8.45-9.05) | 8.78 (8.47-9.08) | 8.41 (8.09-8.72) | 8.72 (8.43-8.98) | 8.04 (7.68-8.38) |
| GPT-4.1 | 8.73 (8.44-9.00) | 8.87 (8.59-9.13) | 8.90 (8.61-9.16) | 8.59 (8.30-8.88) | 8.90 (8.65-9.12) | 8.20 (7.86-8.52) |
| GPT-4o | 9.18 (8.97-9.38) | 9.12 (8.91-9.32) | 9.23 (9.02-9.43) | 9.09 (8.88-9.28) | 8.90 (8.66-9.13) | 8.77 (8.52-9.00) |
| DeepSeek-V3 | 9.23 (9.00-9.44) | 9.20 (9.00-9.39) | 9.10 (8.85-9.33) | 8.79 (8.53-9.02) | 9.13 (8.92-9.32) | 8.76 (8.50-9.00) |
| DeepSeek-V3.1 | 8.90 (8.64-9.13) | 9.05 (8.84-9.24) | 9.02 (8.78-9.25) | 8.41 (8.13-8.68) | 8.99 (8.79-9.18) | 8.44 (8.17-8.70) |
| GPT-4.1 nano | 8.09 (7.77-8.40) | 8.29 (7.99-8.57) | 8.57 (8.28-8.83) | 7.74 (7.42-8.05) | 8.05 (7.74-8.36) | 7.35 (7.00-7.70) |
| Ministral 3B | 7.93 (7.61-8.24) | 8.15 (7.87-8.43) | 8.20 (7.90-8.49) | 7.91 (7.61-8.19) | 7.66 (7.35-7.96) | 7.85 (7.56-8.13) |

To better understand the relative performance of different models across programming languages, Table 7 presents the LLM-judge scores with 95% confidence intervals for each model-language pair.

# D QUALITATIVE EXAMPLES

We start with an example of a Python model completion that did not successfully execute and did not closely resemble the golden completion from the benchmark.

**Example 8: Python Pattern Matching #43**

```python
import matplotlib.pyplot as plt
import numpy as np

class AnalyticsReport:
    def __init__(self, data):
        self.data = data

    def generate_histogram(self, column):
        plt.hist(self.data[column], bins=10, alpha=0.75)
        plt.title(f'Histogram of {column}')
        plt.xlabel(column)
        plt.ylabel('Frequency')
        plt.grid(True)

    def generate_line_chart(self, x_column, y_column):
        plt.plot(self.data[x_column], self.data[y_column], marker='o',
    linestyle='-')
        plt.title(f'Line Chart of {y_column} vs {x_column}')
        plt.xlabel(x_column)
        plt.ylabel(y_column)
        plt.grid(True)
```

**Prefix**

```python
    def generate_bar_chart(self, x_column, y_column):
        plt.bar(self.data[x_column], self.data[y_column], alpha=0.75)
        plt.title(f'Bar Chart of {y_column} vs {x_column}')
        plt.xlabel(x_column)
        plt.ylabel(y_column)
        plt.grid(True)
```

**Golden Completion**

```python
    def generate_scatter_plot(self, x_column, y_column):
        plt.scatter(self.data[x_column], self.data[y_column], alpha=0.75)
        plt.title(f'Scatter Plot of {y_column} vs {x_column}')
        plt.xlabel(x_column)
        plt.ylabel(y_column)
        plt.grid(True)
```

**Model Completion (Claude 3.7 Sonnet)**

```
    def generate_scatter_plot(self, x_column, y_column):
        plt.scatter(self.data[x_column], self.data[y_column], alpha=0.75)
        plt.title(f'Scatter Plot of {y_column} vs {x_column}')
        plt.xlabel(x_column)
        plt.ylabel(y_column)
        plt.grid(True)

# Example usage
if __name__ == '__main__':
    data = {
        'age': np.random.randint(20, 50, size=100),
        'height': np.random.randint(150, 200, size=100),
        'weight': np.random.randint(50, 100, size=100)
    }
    report = AnalyticsReport(data)
    report.generate_histogram('age')
    report.generate_line_chart('height', 'weight')
    plt.show()
```

**Suffix**

```
data = {
    'age': np.random.randint(20, 50, size=100),
    'height': np.random.randint(150, 200, size=100),
    'weight': np.random.randint(50, 100, size=100)
}
report = AnalyticsReport(data)
report.generate_bar_chart('age', 'weight')
assert hasattr(report, 'generate_bar_chart')
assert callable(getattr(report, 'generate_bar_chart'))
assert plt.gcf().get_axes()[-1].get_title() == 'Bar Chart of weight vs
    age'
plt.close('all')
```

**Assertions**

The prefix code shows an AnalyticsReport class with two visualization methods: a histogram and a line chart generator. Both methods follow a consistent pattern of creating a plot, adding a title, labels, and grid. The golden completion provides a generate_bar_chart method that follows the same pattern established in the prefix code. This implementation is crucial for the assertions that test functionality later in the code. The golden completion correctly maintains the structural pattern of the other visualization methods, ensuring consistency in the codebase. Claude 3.7 Sonnet, however, generated a generate_scatter_plot method, which already exists in the suffix, instead of the expected bar chart. This implementation not only deviates from the task of creating a bar chart, but also fails to pass the assertions that specifically test for the presence and functionality of a generate_bar_chart method. While the scatter plot follows the general pattern of the other visualization methods, it represents an incorrect solution to the task. The suffix code shows that the expected implementation should have been a bar chart function, not a scatter plot. This discrepancy reveals the model's failure to properly understand the context and requirements of the task.

Now, we proceed with an example of a Python model completion that successfully executed but did not closely resemble the golden completion from the benchmark.

### Example 9: Python Code2NL/NL2Code #15

```
import requests

def fetch_data_from_api(url):
    """
    Fetches data from the given API URL.

    Parameters:
    url (str): The API endpoint to fetch data from.
```

```
     Returns:
     dict: The JSON response from the API parsed into a dictionary.

     Raises:
     ValueError: If the response status code is not 200.
     """
     response = requests.get(url)
     if response.status_code != 200:
         raise ValueError("API request failed with status code {}".format(
     response.status_code))
     return response.json()

# Example usage:
# data = fetch_data_from_api('https://api.example.com/data')
# print(data)
```

**Prefix**

```
# Call the fetch_data_from_api function with a sample URL and handle the
     response
```

**Golden Completion**

```
def main():
    try:
        data = fetch_data_from_api('https://jsonplaceholder.typicode.com/
    todos/1')
        assert isinstance(data, dict), 'The fetched data should be a
    dictionary'
        print("Data fetched successfully.")
    except ValueError as ve:
        print("Error fetching data:", ve)

    # Additional logic can be placed here

if __name__ == "__main__":
    main()
```

**Model Completion (Claude 3.7 Sonnet)**

```
def main():
    try:
        data = fetch_data_from_api('https://jsonplaceholder.typicode.com/
    todos/1')
        assert isinstance(data, dict), 'The fetched data should be a
    dictionary'
        print("Data fetched successfully.")
    except ValueError as ve:
        print("Error fetching data:", ve)

    # Additional logic can be placed here

if __name__ == "__main__":
    main()
```

**Suffix**

```
assert 'fetches data from the given api url' in fetch_data_from_api.
    __doc__.lower()
assert 'parameters:' in fetch_data_from_api.__doc__.lower()
assert 'returns:' in fetch_data_from_api.__doc__.lower()
assert 'raises:' in fetch_data_from_api.__doc__.lower()
assert 'url (str)' in fetch_data_from_api.__doc__.lower()
```

```
1188  assert 'dict: the json response from the api' in fetch_data_from_api.
1189      __doc__.lower()
1190  assert 'valueerror: if the response status code is not 200' in
1191      fetch_data_from_api.__doc__.lower()
1192  try:
1193      data = fetch_data_from_api('https://jsonplaceholder.typicode.com/
1194      nonexistent')
1195      assert isinstance(data, dict), 'The fetched data should be a
      dictionary'
1196  except ValueError as ve:
1197      assert str(ve) == 'API request failed with status code 404'
```

**Assertions**

The prefix code presents a `fetch_data_from_api` function that retrieves data from a given API URL, with comprehensive documentation including parameters, return values, and potential exceptions. The code also includes commented examples of how to use the function, suggesting a pattern for implementation. The golden completion adds a clear comment that describes what follows: a call to the function with error handling. This simple, descriptive comment serves as NL guidance for the subsequent code block, following good documentation practices. The comment effectively bridges the gap between the function definition and its implementation, making the code more readable and maintainable. This approach aligns with common user behavior observed in telemetry data. Claude 3.7 Sonnet, however, completely duplicated the `main()` function that already exists in the suffix code. While this implementation technically works and passes the assertions, it demonstrates a failure to recognize that the completion should complement rather than duplicate the suffix code. The model has essentially copied the exact content of the suffix rather than providing a meaningful transition between the prefix and suffix.

Now, we proceed with an example of a Python model completion that did not successfully execute but closely resembles the golden completion from the benchmark.

**Example 10: Python Pattern Matching #4**

```
1217  class DataValidator:
1218      def __init__(self, schema):
1219          self.schema = schema
1220
1221      def validate(self, data):
1222          """Validate data against the schema."""
1223          errors = []
1224          for field, field_type in self.schema.items():
1225              if field not in data:
1226                  errors.append(f"Missing field: {field}")
1227              elif not isinstance(data[field], field_type):
1228                  errors.append(f"Incorrect type for field: {field}")
```

**Prefix**

```
1230          else:
1231              if isinstance(data[field], str) and not data[field].strip
1232      ():
1233                  errors.append(f"Empty field: {field}")
```

**Golden Completion**

```
1236          elif not isinstance(data[field], field_type):
1237              errors.append(f"Incorrect type for field: {field}")
```

**Model Completion (DeepSeek-V3)**

```
1240      return errors
1241
      schema = {
```

```
1242        "name": str,
1243        "age": int,
1244        "email": str
1245    }
1246
        validator = DataValidator(schema)
1247    data = {
1248        "name": "John Doe",
1249        "age": 30,
1250        "email": "john.doe@example.com"
1251    }
1252
        errors = validator.validate(data)
1253    assert errors == [], f"Unexpected errors: {errors}"
1254
1255    invalid_data = {
1256        "name": "John Doe",
1257        "age": "thirty",
1258        "email": "john.doe@example.com"
        }
1259
1260    errors = validator.validate(invalid_data)
1261    assert "Incorrect type for field: age" in errors, f"Expected type error
            for age, got: {errors}"
1262    assert len(errors) == 1, f"Expected one error, got: {len(errors)}"
1263
1264    invalid_data = {
1265        "age": 30,
1266        "email": "john.doe@example.com"
        }
1267
1268    errors = validator.validate(invalid_data)
1269    assert "Missing field: name" in errors, f"Expected missing field error
1270        for name, got: {errors}"
1271    assert len(errors) == 1, f"Expected one error, got: {len(errors)}"
```

**Suffix**

```
1274    schema = {
1275        "name": str,
1276        "age": int,
1277        "email": str
        }
1278
1279    validator = DataValidator(schema)
1280    data = {
1281        "name": "",
1282        "age": 30,
1283        "email": "john.doe@example.com"
        }
1284
1285    errors = validator.validate(data)
1286    assert "Empty field: name" in errors, f"Expected empty field error for
1287        name, got: {errors}"
1288    assert len(errors) == 1, f"Expected one error, got: {len(errors)}"
```

**Assertions**

The prefix code presents a DataValidator class that checks data against a schema, validating both field presence and type matching. The validation method accumulates errors in a list and handles two specific validation cases: missing fields and incorrect data types. The golden completion extends the validation logic by adding a third check specifically for string fields, ensuring they aren't empty after stripping whitespace. The golden completion correctly introduces this check as an else branch after the type validation, maintaining the logical flow of the validation

process. DeepSeek-V3, however, duplicated the existing type validation check rather than adding the new empty string validation logic. This duplication creates a logical error, as the same condition (`elif not isinstance(data[field], field_type)`) appears twice in sequence. While DeepSeek-V3's completion structurally resembles the golden completion in that it maintains the pattern of adding conditions related to `data[field]` with appropriate error messages, it fails to introduce the new validation logic needed to pass the assertions in the test suite. The assertion tests specifically verify the ability to detect empty string fields, which the model's completion does not implement. This example demonstrates how a model's completion can closely resemble the golden solution in structure while still containing critical logical errors that prevent proper execution.

# E  ADDITIONAL BENCHMARK AND EXPERIMENTAL DETAILS

## E.1  INFRASTRUCTURE

Our benchmark generation and evaluation workloads were distributed across cloud-based model APIs and local computing resources. Model API calls were orchestrated from a standard laptop (11th Gen Intel i7-1165G7 @ 2.80GHz with 16GB RAM) running Python 3.10. For benchmark generation, using the OpenAI API (GPT-4o) to create synthetic evaluation instances required approximately 2-5 hours of wall-clock time for all languages, depending on API latency and excluding human review time. Each individual model evaluation on the complete benchmark required approximately 1.5-3 hours of wall-clock time, also dependent on the API latency. The execution component of our evaluation pipeline, which verifies functional correctness, was executed on the same laptop and required approximately 15 minutes per model (details in Appendix E.2).

## E.2  FUNCTIONAL CORRECTNESS EVALUATION

Our functional correctness evaluation methodology implements robust, secure, and reproducible execution environments across all six programming languages. Each evaluation instance consists of four components: a context prefix, a golden completion (or model-generated completion during evaluation), a context suffix, and assertion statements that verify correctness. The execution pipeline combines these components into complete, executable programs with language-specific safeguards and dependency management.

**Python Execution Environment.** Python evaluation instances run in controlled subprocesses with 30-second timeouts to prevent infinite loops. We automatically insert `matplotlib` non-interactive backend configuration to prevent `plt.show()` calls from blocking execution, and handle environment variables securely to provide necessary API access while maintaining isolation. When dependency-related errors occur, our system automatically attempts to install missing packages using `pip` before retrying execution. Each evaluation instance runs in its own isolated environment to prevent cross-contamination between tests.

**Java Execution Environment.** Java evaluation uses adaptive compilation strategies based on code complexity. Simple test cases without external dependencies use direct `javac` compilation and execution with assertions enabled via the `-ea` flag. Complex cases requiring external libraries (Apache Commons, Jackson, Guava, etc.) automatically utilize Gradle build management with Maven Central dependency resolution. Our system detects package declarations and import statements to determine the appropriate compilation strategy, ensuring both basic and enterprise-level Java code can be properly evaluated.

**JavaScript and TypeScript Execution.** JavaScript evaluation uses Node.js execution with automatic `npm` package installation for missing dependencies. We configure execution environments with proper PATH resolution to ensure consistent Node.js and npm access across different system configurations. TypeScript evaluation adds a compilation step using `tsc` before JavaScript execution, with automatic installation of TypeScript compiler and type definitions (`@types` packages) as needed. Both environments support up to five retry attempts for dependency resolution to handle multiple missing packages.

**C# Execution Environment.** C# evaluation employs `dotnet run` for basic console applications and full MSBuild project compilation for complex scenarios requiring NuGet packages. Our system automatically detects namespace declarations and external dependencies (Entity Framework,

Newtonsoft.Json, Azure SDKs, etc.) to generate appropriate `.csproj` files with package references. The execution environment targets .NET 6.0 for broad compatibility while supporting modern C# language features.

**C++ Execution Environment.** C++ evaluation uses multiple compiler detection (`g++`, `clang++`) with comprehensive library path resolution for external dependencies. Our system automatically detects common libraries (OpenSSL, Boost, OpenCV, Eigen, etc.) from include statements and configures appropriate compiler flags, include paths, and library linking. We support both Homebrew and system-installed libraries across macOS and Linux platforms, with automatic detection of architecture-specific paths (Apple Silicon vs Intel).

**Cross-Language Safeguards.** All execution environments implement consistent safety measures: isolated temporary directories with automatic cleanup, configurable timeouts (30-60 seconds based on language compilation requirements), comprehensive error handling with detailed diagnostic reporting, and proper resource management to prevent system interference. Dependencies are installed locally within test directories rather than globally to maintain system isolation.

This multi-language execution infrastructure allows us to comprehensively evaluate functional correctness across diverse programming paradigms while maintaining the security and reproducibility essential for reliable benchmarking. The entire evaluation pipeline generates both human-readable reports and structured JSON output for detailed analysis of model performance across languages and categories.

### E.3 BENCHMARK GENERATION PROMPTS

To create **DevBench**'s diverse and realistic evaluation instances, we developed specialized generation prompts that captured the nuances of each programming language and code completion category. These structured prompts guided the GPT-4o model to create evaluation instances that accurately reflect real-world coding scenarios identified in our telemetry analysis. Each prompt was meticulously crafted with specific instructions detailing the characteristic patterns, expected structures, and quality requirements for generating valid evaluation instances. The prompts ensured consistent formatting while maintaining language-specific idioms and patterns, balancing standardization with authentic coding styles. In this section, we present the template prompts used for each language-category pair, demonstrating how we systematically encoded the insights from our telemetry analysis into generative instructions that produced high-quality synthetic evaluation instances while maintaining evaluation instance realism. Due to space constraints, we only include one C++ prompt here; the complete collection of prompts for all languages and categories is available in our code repository.

#### C++: API Usage Prompts

```
API_USAGE_SYSTEM_PROMPT = """
You are an expert C++ developer tasked with creating benchmark examples
    for testing rare API usage and uncommon library function capabilities
     in large language models.
Your role is to generate high-quality, realistic coding scenarios that
    effectively test an LLM's ability to recognize and continue
    established patterns in code involving uncommon APIs and library
    functions.

Your output should be a single JSON object formatted as a JSONL entry.
    The code must be fully executable C++ that passes all assertions.

Key Responsibilities:
1. Generate diverse examples from these API categories (rotate through
    them, don't focus only on file operations or network protocols):
    - Text and font processing (HarfBuzz, FreeType, ICU)
    - Graphics and math libraries (DirectXMath, Eigen, GLM, OpenGL)
    - Security/cryptography APIs (OpenSSL, Botan, Crypto++, wolfSSL)
    - System-level APIs (Windows SDK, POSIX, Linux Kernel, BSD, Mach)
    - Standard libraries (C Standard Library, C++ Standard Library, GNU C
     Library)
    - Web API integration (libcurl, Boost.Beast, cpp-httplib, cpprestsdk)
```

```
      - Machine learning libraries (OpenCV, TensorFlow C++, PyTorch C++,
    ONNX)
      - Cloud services (AWS SDK for C++, Azure SDK for C++, gRPC)
      - Database interfaces (SQLite, MySQL Connector C++, MongoDB C++
    Driver, Redis)
      - File formats and parsing (RapidJSON, nlohmann/json, tinyxml2, yaml-
    cpp)
      - Web frameworks (Drogon, Crow, oatpp, Pistache)
      - Network protocols (Boost.Asio, ZeroMQ, nanomsg)
      - Scientific computing (Eigen, Armadillo, Intel MKL, BLAS, LAPACK)
      - GUI frameworks (Qt, wxWidgets, ImGui, GTK, FLTK)
      - Multimedia (SDL, FFmpeg, OpenAL, libsndfile)
      - Compression (zlib, bzip2, LZMA, LZ4, Zstandard)
      - Cross-platform development (Boost, Qt, wxWidgets)
      - Mobile development (Android NDK, iOS SDK, Core Foundation)
      - Testing frameworks (Google Test, Catch2, Boost.Test)
      - Hardware acceleration (Intel Intrinsics, ARM NEON, CUDA, OpenCL)
      - Legacy/deprecated APIs

2. Ensure patterns are clear and identifiable even with uncommon or
    deprecated APIs
3. Create ground truth completions that represent best practices while
    handling API versioning
4. Write assertions that meaningfully test both API correctness and
    parameter ordering
5. Provide clear justification for why the example makes a good test case
6. Ensure code quality:
      - All code must be fully executable C++
      - All assertions must pass when code is run
      - Include necessary includes and namespaces
      - Handle cleanup of resources
      - Use proper exception handling
      - Include minimal working examples
      - Mock external dependencies where needed
7. Write robust assertions that:
      - Verify actual API behavior
      - Test parameter ordering
      - Check error conditions
      - Validate return values
      - Mock external resources

When generating examples:
1. Focus on less common library functions and domain-specific APIs
2. Test the model's handling of deprecated but valid API patterns
3. Ensure patterns include correct parameter ordering and naming
    conventions
4. Include edge cases in API usage where relevant
5. Keep code focused on demonstrating rare but valid API interactions
"""

API_USAGE_USER_PROMPT = """
You are helping create a benchmark for rare API usage capabilities. Your
    task is to generate a coding scenario that tests an LLM's ability to
    recognize and
complete patterns in C++ code involving uncommon or deprecated APIs.

Generate a single JSONL entry testing rare API usage capabilities. Choose
    from one of these categories (rotate through them, don't focus only
    on file operations or network protocols):
      - Text and font processing (HarfBuzz, FreeType, ICU)
      - Graphics and math libraries (DirectXMath, Eigen, GLM, OpenGL)
      - Security/cryptography APIs (OpenSSL, Botan, Crypto++, wolfSSL)
      - System-level APIs (Windows SDK, POSIX, Linux Kernel, BSD, Mach)
      - Standard libraries (C Standard Library, C++ Standard Library, GNU C
        Library)
```

```
    - Web API integration (libcurl, Boost.Beast, cpp-httplib, cpprestsdk)
    - Machine learning libraries (OpenCV, TensorFlow C++, PyTorch C++,
    ONNX)
    - Cloud services (AWS SDK for C++, Azure SDK for C++, gRPC)
    - Database interfaces (SQLite, MySQL Connector C++, MongoDB C++
    Driver, Redis)
    - File formats and parsing (RapidJSON, nlohmann/json, tinyxml2, yaml-
    cpp)
    - Web frameworks (Drogon, Crow, oatpp, Pistache)
    - Network protocols (Boost.Asio, ZeroMQ, nanomsg)
    - Scientific computing (Eigen, Armadillo, Intel MKL, BLAS, LAPACK)
    - GUI frameworks (Qt, wxWidgets, ImGui, GTK, FLTK)
    - Multimedia (SDL, FFmpeg, OpenAL, libsndfile)
    - Compression (zlib, bzip2, LZMA, LZ4, Zstandard)
    - Cross-platform development (Boost, Qt, wxWidgets)
    - Mobile development (Android NDK, iOS SDK, Core Foundation)
    - Testing frameworks (Google Test, Catch2, Boost.Test)
    - Hardware acceleration (Intel Intrinsics, ARM NEON, CUDA, OpenCL)
    - Legacy/deprecated APIs

CRITICAL JSON FORMATTING REQUIREMENTS:
1. Your response MUST be a syntactically valid JSON object
2. PROPERLY ESCAPE all special characters in strings:
    - Use \\" for double quotes inside strings
    - Use \\n for newlines
    - Use \\t for tabs
    - Use \\\\ for backslashes
3. The entire JSON object must be on a SINGLE LINE
4. Do NOT include formatting or indentation outside the JSON structure
5. DO NOT use markdown code blocks (```) in your response
6. Test your JSON structure before completing your response

Required JSON fields:
- id: A unique numeric identifier
- testsource: Use "synthbench-api-usage"
- language: "cpp"
- prefix: The code that comes before the completion (may or may not
    establish the API pattern)
- suffix: The code that follows the completion (may or may not establish
    the API pattern) - should be DIFFERENT from the golden completion AND
    should include necessary assertions
- golden_completion: The correct API implementation that maintains
    consistency with prefix/suffix and will pass all assertions
- LLM_justification: Explain why this is a good test case and the context
    behind it
- assertions: Leave this field as an empty string - all assertions should
    be integrated into the suffix code

CRITICAL JSON FIELD REQUIREMENTS:
1. ALWAYS include ALL required JSON fields listed above, even if empty
2. The "assertions" field MUST be present with an empty string value: "
    assertions": ""
3. Do NOT omit any fields from your JSON object
4. Format example showing required empty assertions field:
    {"id": "42", ..., "assertions": ""}
5. INCORRECT: {"id": "42", ...} - missing assertions field

CRITICAL CHANGE - NEW SUFFIX REQUIREMENTS:
1. The suffix must contain both execution code AND assertion code
2. Include assert() statements DIRECTLY IN THE SUFFIX at the appropriate
    places
3. All assertions must be placed in the same function/class as the code
    being tested
4. DO NOT create separate assertion functions or classes
5. Place assertions immediately after the code that should be tested
```

```
6. Never duplicate any golden_completion code in the suffix
7. The assertions must pass when the combined prefix + golden_completion
   + suffix is run

Critical Requirements for Avoiding Duplication:
1. The golden_completion field should ONLY contain the solution code that
    fills in the gap
2. The suffix must contain DIFFERENT code that follows after the
    completion
3. Do NOT repeat any golden_completion code in the suffix
4. The suffix field should NEVER duplicate the golden_completion code
5. There should be a clear DISTINCTION between what goes in
    golden_completion vs suffix
6. Ensure clear SEPARATION between completion and suffix content

Include Requirements:
1. Do NOT include headers unless they are ACTUALLY USED in at least one
    of:
   - prefix
   - suffix (including assertions)
   - golden_completion
2. Every included header must serve a clear purpose
3. Do not include "just in case" headers that aren't used
4. All required includes must appear in the prefix section
5. If an include is only needed for the golden_completion, it must still
    appear in the prefix
6. Make sure to include <cassert> header for assert() statements

PREFIX LENGTH REQUIREMENTS - CRITICAL:
1. The PREFIX section MUST be SUBSTANTIALLY LONGER than other sections
2. The prefix MUST be AT LEAST 50-60 lines of code - this is an absolute
    requirement
3. Provide extensive context and setup code in the prefix
4. Include helper functions, utility classes, and related code structures
5. Add detailed comments and explanations within the prefix
6. The prefix should demonstrate a comprehensive but incomplete
    implementation
7. Add relevant constants, configuration objects, and data structure
    initialization

Indentation requirements:
1. All code sections must maintain consistent indentation
2. If code is inside a function/class:
- The prefix should establish the correct indentation level
- The golden_completion must match the prefix's indentation
- The suffix must maintain the same indentation context
- Assertions should be at the appropriate scope level
3. Ensure proper dedenting when exiting blocks
4. All code blocks must be properly closed

The API pattern can be established either in the prefix or suffix code.
The golden completion should demonstrate understanding and correct usage
    of the API pattern regardless of where it is established.

Code requirements:
1. Must be fully executable C++ code
2. All assertions must pass when run
3. Include all necessary headers and namespaces
4. Mock external dependencies
5. Clean up resources properly
6. Handle errors appropriately
7. Assertions must be placed BEFORE cleanup code
8. Resource cleanup must be in the suffix AFTER all assertions
9. All assertions must complete before any cleanup occurs
```

```
CRITICAL CODE STRUCTURE REQUIREMENTS:
1. NEVER place code outside of functions or classes
2. ALL code must be contained within proper C++ scope boundaries
3. DO NOT place assertions or standalone code statements at the global/
    namespace level
4. ALL assertions must be contained within functions (such as main() or
    other functions)
5. ALWAYS ensure code is properly nested within appropriate class and
    function structures
6. NEVER generate code that would compile as a partial class
7. NEVER duplicate class definitions – each class must be defined only
    once
8. Verify that the beginning and end of classes and functions are
    properly matched with braces {}
9. DO NOT leave any code statements outside of function bodies
10. Place all assertions within appropriate functions (main(), test(),
    etc.)

CRITICAL ASSERTION PLACEMENT:
1. All assert() statements must be placed DIRECTLY IN THE SUFFIX code
2. Assertions should be placed immediately after the code that needs to
    be verified
3. Assertions must be within the same function as the code being tested
4. Assertions must be executed BEFORE any cleanup code
5. Assertions must be properly indented to match the surrounding code
    structure
6. Use assert(condition) format for all assertions
7. Make sure <cassert> is included for assert() statements

Requirements:
1. The scenario should demonstrate a clear pattern recognizable with the
    given context
2. The completion section should focus on rare library functions
3. The pattern should follow correct API conventions across different
    versions
4. Ground truth should demonstrate proper parameter ordering
5. Assertions should verify API behavior and parameter correctness
6. Include comments indicating API version compatibility and parameter
    requirements

Format your response as a single line JSON object with newlines escaped
    appropriately.

Example format:
{"id": "1", "testsource": "synthbench-api-usage", "language": "cpp", "
    prefix": "...", "suffix": "...", "golden_completion": "...", "
    LLM_justification": "...", "assertions": "..."}

VALIDATION CHECKLIST BEFORE SUBMITTING:
1. Have you properly escaped ALL special characters?
2. Is your entire response a single, valid JSON object?
3. Are all string values properly quoted and terminated?
4. Have you verified there are no unescaped newlines in your strings?
5. Have you checked for balanced quotes and braces?
6. Is your prefix at least 50-60 lines of code?
7. Have you used clear distinctions between golden_completion and suffix?
8. Have you included all assertions DIRECTLY IN THE SUFFIX code?
9. Have you verified that assertions will pass when the code is executed?
10. Is the assertions field included with an empty string value ("
    assertions": "")?
11. Have you verified that ALL required fields are present in your JSON?
12. Have you verified your example is NOT one of the prohibited trivial
    examples?
13. Does your example meet ALL the complexity validation criteria?
14. Does your example demonstrate genuinely advanced C++ features?
```

Table 8: Similarity metrics across task categories. All reported results used a temperature of 0.2.

| Model | Average Cosine Similarity | | | | | | Line 0 Exact Match Rate (%) | | | | | |
|---|---|---|---|---|---|---|---|---|---|---|---|---|
| | API Usage | Code2NL NL2Code | Purpose Underst. | Low Context | Pattern Matching | Syntax Compl. | API Usage | Code2NL NL2Code | Purpose Underst. | Low Context | Pattern Matching | Syntax Compl. |
| Claude 4 Sonnet | 0.57 | 0.58 | 0.71 | 0.75 | 0.66 | 0.61 | 36.33 | 52.0 | 57.33 | 62.0 | 53.67 | 47.33 |
| Claude 3.7 Sonnet | 0.50 | 0.59 | 0.65 | 0.75 | 0.70 | 0.59 | 31.67 | 53.0 | 52.33 | 62.33 | 53.0 | 44.33 |
| GPT-4.1 mini | 0.57 | 0.49 | 0.72 | 0.79 | 0.67 | 0.61 | 37.0 | 44.33 | 56.67 | 64.33 | 52.0 | 45.33 |
| GPT-4.1 | 0.57 | 0.48 | 0.69 | 0.77 | 0.70 | 0.62 | 38.33 | 43.0 | 52.33 | 63.0 | 53.0 | 46.0 |
| GPT-4o | 0.56 | 0.48 | 0.72 | 0.77 | 0.70 | 0.63 | 37.33 | 42.33 | 56.0 | 64.0 | 54.33 | 47.0 |
| DeepSeek-V3 | 0.58 | 0.53 | 0.71 | 0.79 | 0.75 | 0.65 | 37.33 | 47.33 | 54.0 | 62.67 | 60.0 | 48.67 |
| DeepSeek-V3.1 | 0.56 | 0.50 | 0.68 | 0.77 | 0.69 | 0.62 | 39.67 | 48.33 | 56.0 | 61.67 | 57.33 | 50.0 |
| GPT-4.1 nano | 0.49 | 0.39 | 0.64 | 0.71 | 0.59 | 0.53 | 25.33 | 32.67 | 45.67 | 51.33 | 41.67 | 34.33 |
| Ministral 3B | 0.41 | 0.36 | 0.53 | 0.55 | 0.53 | 0.43 | 20.67 | 27.0 | 35.33 | 33.67 | 29.67 | 23.67 |

```
Important:
- Never place cleanup code before assertions
- Keep all verification code before any cleanup
- Ensure resources exist when assertions run
- Use proper try/finally blocks if needed
- Maintain correct execution order
- ALL ASSERTIONS SHOULD BE IN THE SUFFIX, not in a separate assertions
    field

Ensure the example is self-contained and can be evaluated independently.
    All assertions must pass when run.
Use proper escaping for newlines/quotes and maintain indentation in the
    escaped strings.

"""
```

### E.4   EVALUATION PROMPT

For our model evaluation process, we implemented a carefully designed prompt template focused on precise code completion tasks. After initial experimentation revealed that different prompt formats could significantly impact model performance, including Claude 3.7 Sonnet, due to formatting issues, we selected a structured instruction-based approach that addresses common failure modes. Our code repository contains the full evaluation prompt.

The selected prompt format provides clear examples demonstrating proper replacement behavior in various scenarios, explicitly instructing models to maintain correct indentation and avoid duplicating existing code structures. By standardizing the input format with clear `#TODO: You Code Here markers` and providing explicit instructions against common mistakes, we created a more level evaluation environment that better isolates models' code understanding capabilities from prompt interpretation abilities.

This evaluation prompt design aligns with real-world code completion scenarios where maintaining contextual formatting is essential for functional correctness, ensuring our benchmark more accurately reflects models' practical utility in development environments. Performance differences observed between models using this standardized prompt more reliably indicate their intrinsic code completion capabilities rather than their ability to navigate ambiguous or unstructured prompting patterns.

### E.5   FULL SIMILARITY METRICS BY CATEGORY

We provide the complete similarity-based results across all evaluated models in Table 8. This expanded view offers a more comprehensive comparison of model performance across different categories and similarity dimensions.

Table 9: Pass@1 rates by programming language using $n = 5$ samples. All results used a temperature of 0.2.

| Model | Python | JavaScript | TypeScript | Java | C++ | C# |
|---|---|---|---|---|---|---|
| Claude 4 Sonnet | 74.5% | 79.6% | 78.9% | 93.1% | 93.7% | 88.9% |
| Claude 3.7 Sonnet | 73.9% | 77.3% | 71.9% | 88.4% | 88.4% | 83.5% |
| GPT-4.1 mini | 72.1% | 78.1% | 67.3% | 87.5% | 87.7% | 85.5% |
| GPT-4.1 | 70.1% | 77.1% | 68.3% | 88.0% | 86.3% | 82.2% |
| GPT-4o | 70.3% | 77.3% | 65.2% | 84.3% | 84.7% | 81.4% |
| DeepSeek-V3 | 72.7% | 71.3% | 65.2% | 85.7% | 77.8% | 79.2% |
| DeepSeek-V3.1 | 64.9% | 67.5% | 64.7% | 78.7% | 78.3% | 78.0% |
| GPT-4.1 nano | 57.5% | 65.4% | 58.7% | 76.0% | 71.7% | 69.1% |
| Ministral 3B | 34.4% | 50.9% | 41.8% | 53.7% | 58.0% | 53.1% |

### E.6 PASS@1 PERFORMANCE BY PROGRAMMING LANGUAGE

Table 9 provides a detailed breakdown of Pass@1 performance across all six programming languages for each evaluated model. These results reveal language-specific strengths and weaknesses that complement the overall performance metrics reported in the main paper.

## F  LIMITATIONS AND FUTURE DIRECTIONS

While **DevBench** represents a significant advancement in code generation evaluation, we identify several opportunities for future enhancement and extension.

### F.1  EXPANDING BENCHMARK GENERATION DIVERSITY

Our synthetic, telemetry-driven generation approach effectively prevents data contamination and limits bias by leveraging GPT-4o as the generation model. To further enhance diversity, future iterations could incorporate multiple foundation models with varied training backgrounds. This approach will maintain our telemetry-driven, human-validated methodology while expanding stylistic diversity.

The derivation of test categories from real-world telemetry data grounds our benchmark in authentic developer experiences. Building on this foundation, future research could explore federated learning approaches that enable even closer alignment with real developer interactions while maintaining privacy safeguards.

### F.2  ENHANCING EVALUATION FRAMEWORKS

The complementary evaluation metrics we employ (Pass@1, similarity-based metrics, and LLM-judge assessments) provide multidimensional insights into model performance. The occasional divergence between these metrics—such as cases where higher syntactic similarity does not correlate with functional correctness—highlights an opportunity to develop composite metrics that better capture the full spectrum of code quality dimensions relevant to developers.

Our LLM-judge uses o3-mini as the scoring model, selected for its favorable bias profile as documented in the OpenAI System Card showing lowest bias on discrimination tasks (OpenAI et al., 2025c). Future work could explore ensemble judging approaches, human-in-the-loop calibration, or contrastive evaluation techniques that specifically control for stylistic biases, allowing for even more robust evaluation.

### F.3  BROADENING COVERAGE SCOPE

**DevBench** currently provides strong coverage of code completion scenarios while offering opportunities to expand into additional development activities. Future extensions could apply our methodology

to generate synthetic evaluation instances for code refactoring, debugging, multi-file architecture design, and system-level programming challenges—further enriching the evaluation landscape.

Our language coverage, which already includes six major programming languages (Python, JavaScript, TypeScript, Java, C++, and C#), provides a foundation for expansion. Future iterations could incorporate emerging languages such as Rust, Go, Ruby, and Swift, as well as develop more complex, multi-stage evaluation instances that reflect the challenges of professional software engineering.

### F.4 OPTIMIZING RESOURCE EFFICIENCY

The benchmark generation process, while relatively affordable using current API pricing (approximately \$5.00/1M input tokens and \$20.00/1M output tokens for GPT-4o (OpenAI, 2025)), presents opportunities for further efficiency improvements. Future work could provide streamlined tools and templates for benchmark extension, reducing the expertise required to create custom evaluation instances while maintaining quality standards.

Response latency represents another dimension deserving further exploration, as it can impact developer workflow and productivity. Incorporating systematic latency evaluation alongside quality metrics would provide a more holistic view of the practical trade-offs involved in model selection.

### F.5 ADVANCING FAIRNESS AND INCLUSIVITY

The telemetry data that informs our benchmark categories derives from diverse developer interactions, offering an opportunity to explicitly analyze potential implicit biases in programming styles, paradigms, or practices. Future research could conduct systematic analyses of representation across different programming communities and traditions, ensuring the benchmark remains equitable and inclusive.

Performance disparities across programming languages present another avenue for methodological refinement. Future extensions could develop language-specific normalization techniques or targeted improvements for underrepresented languages, ensuring fairness across diverse developer communities and technical ecosystems.

## G BROADER IMPACTS

Our **DevBench** benchmark has several potential positive societal impacts. By enabling more accurate evaluation of code completion models, our work can lead to improved developer productivity tools that reduce repetitive coding tasks, decrease the time required to implement software solutions, and potentially lower barriers to entry in programming by assisting novice developers. More accurate code completion could also improve software quality by suggesting well-tested patterns and reducing common programming errors, potentially leading to more reliable and secure software systems.

However, we also acknowledge several potential negative impacts. First, there are fairness considerations related to programming language representation; our benchmark's coverage of six languages, while broader than many existing benchmarks, still represents a limited subset of the programming ecosystem. This may lead to uneven improvements across programming languages, potentially disadvantaging developers who work primarily with languages not included in our benchmark. Second, there are potential job market implications if increasingly capable code completion systems begin to automate significant portions of software development tasks, potentially affecting employment opportunities for certain types of programming roles.

Additionally, we recognize that improvements in code generation capabilities could have security implications. While our benchmark focuses on code completion rather than full program generation, advances in code synthesis could potentially be misused to generate malicious code more efficiently or to exploit vulnerabilities in existing systems. To mitigate these concerns, we have designed our benchmark to emphasize proper API usage, security patterns, and code quality metrics rather than merely measuring functional correctness.

To address these concerns, we have made our benchmark and methodology publicly available to enable community scrutiny, external validation, and continuous improvement. We encourage future

research to extend language coverage, develop more diverse evaluation metrics, and carefully monitor potential misuses of increasingly capable code generation systems.

