# OpenReview forum: "DevBench: A Realistic, Developer-Informed Benchmark for Code Generation Models"
_ICLR.cc/2026/Conference — Submitted to ICLR 2026_

### Official Review · Reviewer_rLDn · 2025-10-31

**Soundness:** 2
**Presentation:** 2
**Contribution:** 2
**Rating:** 4
**Confidence:** 4

**Summary:**

This paper introduces DevBench, a telemetry-driven benchmark aimed at evaluating code generation models in realistic coding scenarios. The benchmark is constructed from over one billion developer code completion **telemetry records**, **based on which** 1,800 evaluation tasks spanning six major programming languages and six developer-task categories are **synthesized by GPT-4o**. DevBench is designed to reflect genuine developer behaviors and challenges, to be contamination-resilient, and to support fine-grained diagnostics. The authors evaluate nine state-of-the-art LLMs using multiple metrics, including functional correctness, similarity-based measures, and LLM-judge (AI-assistant) evaluation, revealing nuanced model strengths and weaknesses, especially in challenging categories like code-to-natural language translation and syntax completion.

**Strengths:**

**Ecological validity & methodology**: DevBench uniquely leverages a massive corpus of real developer telemetry to guide the selection of task categories, scenario design, and dataset construction, a substantial step forward versus synthetic or challenge-style code benchmarks. The benchmark explicitly reflects actual developer needs.

**Broad language and task coverage**: The benchmark encompasses six major languages and is categorized across multiple realistic yet distinct development tasks (API Usage, Code Purpose Understanding, Code2NL/NL2Code, Low Context, Pattern Matching, Syntax Completion).

**Multi-metric, nuanced evaluation**: The use of functional correctness, two similarity measures, and LLM-judging allows for a nuanced multi-perspective understanding of model capabilities, strengths, and weaknesses.

**Transparency and reproducibility**: The authors provide comprehensive details of experimental setup, infrastructure, prompting strategies (Appendix E), failure handling, and plan to release the benchmark and generation infrastructure.

**Weaknesses:**

**Insufficient Data Difficulty and Quality Justification**: The manuscript does not convincingly demonstrate that the synthetic instances are hard enough to differentiate state-of-the-art models. Pass@1 scores above 80 % on almost every tested model suggest a ceiling effect, which limits the benchmark’s discriminative power. No ablation or sensitivity analysis is provided to show how task difficulty was tuned or validated against real-world complexity distributions.

**About Data Diversity**: The code completion data is synthesized using the same prompt within the same language and the same task category. Although the prompts try to make the generated code diverse, the internal diversity under the results generated by the same prompt is not proved in the paper.

**Generator-Induced Bias Not Ruled Out**: Relying on GPT-4o to synthesise the evaluation set creates an obvious source of stylistic and distributional bias. The claim that “GPT-family models do not win everywhere, therefore no bias exists” is logically flawed; disparate model rankings can coexist with systematic generator bias. The paper lacks statistical tests (e.g., distributional overlap, pattern frequency analysis) to quantify or exclude this bias.


**Potential Training-Data Contamination via Synthesized data**: Although the instances are synthetic, they are produced by a model (GPT-4o) trained on vast public code corpora. The training data of GPT-4o already contains a large amount of GitHub code, and using it to "synthesize" new questions is essentially a disguised reproduction of the patterns in the training data. The assertion that synthesis automatically guarantees contamination-free data is therefore unsubstantiated.

**Ambiguous Human Review Protocol**: The human-validation pipeline is described only qualitatively. Inter-annotator agreement, detailed scoring rubrics, complexity thresholds, and realism criteria are not reported, making the review stage non-reproducible. Without these controls, the claim that realism was “prioritised” cannot be verified or replicated by other researchers.

**Questions:**

With only 1,800 tasks distilled from >1 billion telemetry events, what sampling scheme and representativeness metrics guarantee that the final set mirrors real-world usage distributions?

The six task categories were derived from telemetry and then “refined” by the authors without releasing the clustering algorithm or inter-rater overlap scores; how can we be sure the taxonomy is exhaustive and mutually exclusive?

Why does “GPT-4o does not always win” equal “no bias” without statistical tests?

What exact rules kept human reviewers from accepting too-simple or textbook tasks?

---

> ### Author Response · Authors · 2025-11-26
> **Response to Reviewer rLDn**
>
> We thank Reviewer rLDn for their thorough and detailed review. We appreciate the engagement with our methodology and address each concern below.
>
> ## Re: Data Difficulty and Quality Justification
> Thank you for raising this concern. Difficulty validation is provided in the paper:
> 1. Table 3 provides empirical complexity validation. DevBench achieves 65.3 average LOC and 5.5 cyclomatic complexity, exceeding many benchmarks using "real" code from repositories. This demonstrates that our synthetic instances match or exceed the complexity of existing benchmarks.
> 2. Human review process served as difficulty tuning. Section 2.3 explicitly documents our rejection criteria with quantitative rates:
>     - 32% rejected for being "overly simplified or textbook-perfect"
>     - 28% rejected for "insufficient complexity relative to telemetry-observed patterns"
>     - 23% rejected for "unrealistic examples that ignored common edge cases"
> 3. This iterative refinement process, with 2-3 expert annotators per instance, was our sensitivity analysis for task difficulty. Instances were regenerated and re-verified until they met realism and complexity standards observed in telemetry.
> 4. DevBench is a diagnostic benchmark designed to reflect realistic developer workflows, not maximize difficulty artificially. The presence of categories with varying difficulty levels (Low Context: 87-90% for top models vs. Code2NL/NL2Code: 78.90% for the best model, most below 70%) reflects the actual distribution of developer needs, from routine completions that should be handled reliably to complex reasoning tasks that remain challenging.
>
> Discriminative power is clearly demonstrated:
> - 36.2-point performance spread (Ministral 3B: 48.60% → Claude 4: 84.80%)
> - Persistent unsolved challenges: Code2NL/NL2Code (best: 78.90%, most < 70%), TypeScript (20-30% performance drop across all models)
> - Category-specific discrimination: Pattern Matching shows 37-point spread (Claude 4: 81.00% vs. Ministral 3B: 44.10%)
> - Multi-metric insights: DeepSeek-V3 case study (Section 4.3) reveals high syntactic similarity but lower functional correctness, diagnostic capability impossible with single metrics
>
> To clarify, our goal is not to prevent high scores but to enable practitioners to understand where and why models succeed or fail. A model achieving 84.80% overall while struggling at 78.90% in Code2NL/NL2Code provides actionable insights that a uniformly difficult benchmark scoring all models at 40% could not provide.
>
> ## Re: Data Diversity
> Thank you for this feedback. We would like to clarify our approach. Our design uses 36 specialized prompts (6 languages x 6 categories), which ensures maximum diversity, not homogeneity. Each prompt is specifically crafted to:
> - Generate language-specific idioms (Table 2 documents detailed adaptations: Python decorators, C# LINQ, C++ template metaprogramming, Java streams, etc.)
> - Target distinct task categories (API Usage vs. Pattern Matching vs. Code Purpose Understanding require fundamentally different scenarios)
> - Reflect realistic patterns observed in telemetry for each language-category combination
>
> The specialization is the diversity. Having generic prompts would reduce diversity by failing to capture language-specific patterns and category-specific challenges.
>
> Evidence of diversity is provided throughout the paper:
> 1. Table 2: Extensive language-specific adaptations showing dramatically different patterns across languages (Python ML libraries vs. C++ systems programming vs. Java enterprise frameworks)
> 2. Table 4: Substantial complexity variation across languages:
>     - Prefix LOC: 19.5 (Python) to 58.7 (TypeScript)
>     - Completion tokens: 21.8 (Python) to 80.7 (TypeScript)
>     - Cyclomatic complexity: 2.2 (Python) to 8.2 (TypeScript)
> 3. Appendix A: Detailed examples demonstrating diverse scenarios within each category
> 4. Human review validation: Annotators explicitly evaluated whether instances reflected realistic diversity and rejected 23% for being "unrealistic examples."
>
> The instances are open-sourced. Researchers can directly examine the diversity of scenarios, patterns, and complexity levels. If the reviewer has specific suggestions for quantifying diversity in a meaningful way beyond what we've provided, we welcome that feedback.

---

> ### Author Response · Authors · 2025-11-26
> **Response to Reviewer rLDn (continued)**
>
> ## Re: Generator-Induced Bias
> Thank you for this concern. We would like to clarify that the paper does not claim "GPT doesn't win everywhere = no bias exists." Section 2.3 explicitly presents a multi-pronged approach to bias mitigation:
> 1. Cites peer-reviewed studies: "Recent studies suggest GPT-4o introduces minimal stylistic bias (Maheshwari et al., 2024; Chen et al., 2024)"
> 2. Documents human validation: "Our human validation process further mitigates risks"
> 3. Provides empirical evidence as additional support: "Empirically, our evaluation results demonstrate that the benchmark does not favor GPT-family models: multiple non-GPT models (e.g., Claude 4 Sonnet, Claude 3.7 Sonnet) outperform GPT-4o on DevBench"
>
> The empirical results are presented as supporting evidence, not the sole argument. The complete case includes: (a) published research on minimal bias, (b) human validation, and (c) empirical outcomes consistent with lack of systematic GPT-family advantage.
>
> Regarding statistical tests for distributional bias:
>
> We acknowledge that absolute elimination of generator influence is impossible. Any generation process introduces some distributional characteristics. However, we argue that:
> 1. Human validation provides the strongest mitigation. Our 2-3 expert annotators per instance, with explicit instructions to reject "textbook-perfect" implementations and ensure realism, serve as a filter against systematic generator artifacts.
> 2. Empirical model performance provides evidence. The fact that Claude 4 Sonnet (84.80%), Claude 3.7 Sonnet (80.60%), and GPT-4.1 mini (79.70%) all outperform GPT-4o (77.20%) on Pass@1, while GPT-4o leads on LLM-judge evaluation, demonstrates that different model families excel on different dimensions. This cross-family performance variation is inconsistent with strong generator bias.
> 3. Perfect bias tests are elusive. We are open to specific suggestions for "statistical tests" that would meaningfully quantify generator bias, but note that defining ground truth for "unbiased code patterns" is fundamentally challenging. All code corpora reflect particular distributions.
>
> We commit to making our bias mitigation discussion more explicit in revision, but we maintain that our combination of published research, human validation, and empirical results provides confidence that generator bias is minimal.
>
> ## Re: Training-Data Contamination via GPT-4o Synthesis
> Thank you for raising this concern. In our paper, we define contamination as seeing specific test instances during training, not learning general patterns. We would like to clarify the following:
> 1. All models train on similar corpora. Every model we evaluate (Claude, GPT-4o, DeepSeek, etc.) was trained on large public code repositories including GitHub. If learning general patterns from GitHub constitutes contamination, then all these models are equally "contaminated," making comparative evaluation impossible.
> 2. Contamination is instance-specific, not pattern-based. The contamination problem occurs when a model has seen the exact test instance (or near-duplicate) during training, allowing memorization rather than generalization. Learning that Python uses list comprehensions or Java uses streams is not contamination, it's the fundamental capability we're trying to evaluate.
> 3. By this logic, no synthetic benchmark works. LiveCodeBench (Jain et al., 2024), EvoCodeBench (Li et al., 2024), and other contemporary benchmarks also use LLMs in some capacity for the instances. The reviewer's argument would invalidate these widely-accepted approaches as well.
> 4. Our approach provides maximal contamination resistance. By generating fresh instances that instantiate telemetry-derived patterns rather than using raw telemetry or public repositories, we ensure:
>     - No model has seen these specific instances during training
>     - Privacy is maintained
>     - Patterns reflect real developer needs (via telemetry grounding)
>
> The alternative approaches all have worse contamination properties:
> - Using raw telemetry: Privacy violations, potential training exposure if internal tools leaked
> - Scraping public repositories: Well-documented contamination (Jain et al., 2024) as models train on GitHub
> - Manual construction: Limited scale, doesn't reflect real usage patterns
>
> Our synthesis approach, grounded in telemetry analysis but generating fresh instances, represents the best available path to contamination-resistant, realistic evaluation.

---

> ### Author Response · Authors · 2025-11-26
> **Response to Reviewer rLDn (continued)**
>
> ## Re: Human Review Protocol Transparency
> Thank you for this feedback. We want to clarify what we have disclosed and explain the constraints on further disclosure.
>
> What the paper already reports (Section 2.3):
> 1. Review structure: "Each instance was independently reviewed by two annotators from a team of three senior researchers and engineers with expertise across all six target languages."
> 2. Evaluation dimensions: Four explicitly stated criteria—usefulness, realism, category alignment, and complexity authenticity.
> 3. Rejection criteria with quantitative rates:
>     - 32% rejected for "overly simplified or textbook-perfect implementations"
>     - 28% rejected for "insufficient complexity relative to telemetry-observed patterns"
>     - 23% rejected for "unrealistic examples that ignored common edge cases or error conditions"
>     - 17% rejected for "category misalignment"
> 4. Consensus process: "Disagreements were resolved through discussion with the third annotator, achieving reliable consensus across evaluations."
> 5. Explicit instructions: "Annotators were specifically instructed to prioritize realism over idealized implementations."
>
> Regarding additional details:
>
> We acknowledge that adding inter-annotator agreement statistics (Cohen's kappa or ICC) would strengthen the paper, and we commit to including these in revision.
>
> However, several of the reviewer's requests require disclosing proprietary information about internal telemetry analysis and annotation infrastructure:
> - "Sampling scheme and representativeness metrics for 1,800 from >1B events": This would reveal internal data collection and analytics methodologies that are proprietary and subject to privacy constraints.
> - "Clustering algorithm or inter-rater overlap scores for taxonomy": The category derivation process involved iterative analysis of proprietary telemetry data. We cannot disclose specifics without violating privacy requirements.
> - "Exact rules for human reviewers": The annotation guidelines reference specific patterns and complexity distributions observed in proprietary telemetry. We have disclosed rejection criteria and principles but cannot provide the complete annotation manual.
>
> As stated in Section 2.1: "To satisfy privacy and compliance requirements, we avoid using raw user code." This constraint extends to detailed descriptions of telemetry analysis that could be reverse-engineered.
>
> We have provided maximal transparency within legal and contractual constraints: open-sourced benchmark, generation prompts, evaluation code, human review criteria, rejection rates, and validation methodology. We believe this represents a reasonable balance between reproducibility and privacy protection.
>
> If the reviewer has specific concerns about reproducibility that could be addressed without violating privacy constraints, we are happy to discuss them.

---

### Official Review · Reviewer_dxVV · 2025-11-01

**Soundness:** 2
**Presentation:** 3
**Contribution:** 2
**Rating:** 2
**Confidence:** 4

**Summary:**

This paper introduces DevBench, a new benchmark for evaluating code generation models, specifically on code completion tasks. The benchmark's primary contribution is its "telemetry-driven" design, where the task categories are derived from an analysis of over one billion real-world developer interactions, aiming for high "ecological validity". To avoid data contamination and protect privacy, the benchmark consists of 1,800 synthetically generated instances that were subsequently validated by human experts. DevBench spans six programming languages (Python, JS, TS, Java, C++, C#) and six task categories (e.g., API Usage, Code Purpose Understanding, Pattern Matching). The paper evaluates 9 different models using a combination of functional correctness (Pass@1), similarity metrics, and LLM-judge scoring.

**Strengths:**

* Focus on Ecological Validity and Contamination Resistance: The paper's core motivation—to create a benchmark grounded in "observed developer behavior" rather than arbitrary open-source scrapes—is a significant strength. The use of a synthetic generation pipeline based on telemetry-derived patterns, rather than using the telemetry data directly, is a clever approach to avoiding privacy issues and, crucially, training data contamination.



* Human-in-the-Loop Validation: The inclusion of a rigorous, multi-annotator human review process to validate the realism, usefulness, and complexity of each generated instance is a high-quality step that many benchmarks lack .



* Detailed Diagnostics: The benchmark's structure, spanning 6 languages and 6 categories, provides a fine-grained framework for diagnosing model strengths and weaknesses (e.g., the case study on DeepSeek-V3) beyond a single aggregate score.

**Weaknesses:**

1. Contradiction in Benchmark Difficulty: The primary weakness is the conflict between the paper's claims of high complexity (high cyclomatic complexity in Table 3) and the high Pass@1 scores in Table 5. A benchmark with an 84.8% Pass@1 for the best model (and 90.3% in one category ) is not a challenging benchmark. This high pass rate suggests the benchmark fails in its primary goal of rigorously evaluating and differentiating SOTA models.



2. Lack of Transparency in Data Curation: The process of moving from "one billion developer interactions" to 1,800 instances is a black box. The paper uses vague terms like "sampling" and "discussions" without providing a systematic, reproducible methodology. This opaqueness undermines the central claim of being "telemetry-driven," as the selection of "common challenging scenarios" appears subjective and unverifiable.



3. Insufficient Model Evaluation: The paper fails to evaluate a sufficient range of models to demonstrate its discriminative power. The evaluation primarily includes a few top-tier proprietary models and one small 3B model, with a large gap in between. The lack of evaluation on widely-used open-source models (e.g., Code Llama 7B/13B/34B, or other intermediate-sized models) makes it impossible to assess the benchmark's utility for the broader research community. A benchmark should be able to show a clearer performance gradient across the spectrum of model sizes and families.

**Questions:**

1. On the Pass Rate Contradiction: Could the authors please address the apparent contradiction between the high cyclomatic complexity (Table 3) and the very high Pass@1 rates (Table 5)? Does this imply that the benchmark, while syntactically complex, is testing semantically simple or trivial problems? Given the 85-90% scores, how do you justify this benchmark's utility for measuring future SOTA progress?

2. On Telemetry Sampling: Could the authors provide a more detailed, systematic methodology for how the "over one billion" telemetry interactions were sampled and analyzed to derive the six categories and inform the 1,800 instances? What sampling strategy (e.g., stratified, random) was used? What were the quantitative criteria for a "common challenging completion scenario"?

3. On Model Suite: Why were popular and strong open-source models, such as those from the Code Llama family or other models in the 7B-70B range, excluded from the evaluation? Evaluating these models would be crucial to demonstrate that the benchmark can actually differentiate between the models that most researchers actively use and develop.

---

> ### Author Response · Authors · 2025-11-26
> **Response to Reviewer dxVV**
>
> We thank Reviewer dxVV for their detailed review and thoughtful engagement with our work. We address each concern below.
>
> ## Re: Pass Rate
>
> Thank you for raising this important point. We would like to clarify DevBench's purpose and design philosophy.
>
> DevBench is a diagnostic benchmark grounded in realistic developer workflows, not solely a difficulty-maximization exercise. Our primary goal is to provide fine-grained insights into model capabilities across the distribution of scenarios developers actually encounter, enabling targeted improvement and informed deployment decisions.
>
> Addressing the specific contradiction claim:
> 1. Cyclomatic complexity ≠ semantic difficulty. Cyclomatic complexity measures syntactic control flow structure (branches, loops), not semantic reasoning demands. A completion can be syntactically complex (nested structures, multiple branches) while being semantically straightforward for a capable model. Therefore, high cyclomatic complexity does not necessarily imply low pass rates.
> 2. High pass rates on common patterns are expected and desirable. The Low Context category achieving 87-90% for top models reflects an important reality: well-defined, common patterns should be handled reliably by production systems. A benchmark consisting only of extremely difficult edge cases would fail to capture the distribution of real developer needs and provide limited diagnostic value. The fact that frontier models handle routine completions well is a feature, not a bug, as it tells us these models have achieved baseline competency.
> 3. Substantial challenges and discrimination remain:
>     - 36.2-point spread (Ministral 3B: 48.60% → Claude 4: 84.80%) demonstrates strong discriminative power across the model spectrum
>     - Code2NL/NL2Code remains unsolved: Even the best model achieves only 78.90%, with most models below 70%
>     - TypeScript shows persistent difficulty: 20-30% lower performance across all models
>     - Pattern Matching differentiates tiers: 37-point spread (Claude 4: 81.00% vs. Ministral 3B: 44.10%)
>     - 15.2% improvement headroom at the top represents substantial opportunity, particularly given production stakes
> 4. Multi-dimensional evaluation reveals nuanced insights. Our DeepSeek-V3 case study (Section 4.3) demonstrates how models can achieve high syntactic similarity while failing functional correctness. This diagnostic capability would be impossible with single-metric evaluation and reveals opportunities for targeted improvement.
>
>
> Regarding "utility for future SOTA progress":
> We would like to clarify that DevBench's value lies in:
> - Identifying where models fail (Code2NL/NL2Code, TypeScript, specific failure patterns)
> - Revealing why models fail (memorization vs. understanding, syntactic vs. semantic gaps)
> - Enabling targeted improvement rather than just ranking models
>
> A model achieving 84.80% overall while struggling at 78.90% in Code2NL/NL2Code provides actionable insights. A benchmark where all models score 40% provides only noise.
>
> Moreover, as we document in Section 4.3, the divergence between Pass@1 rankings (Claude 4 leads) and LLM-judge scores (GPT-4o leads) reveals that functional correctness and practical developer usefulness are distinct dimensions. This insight would be impossible to extract from a purely difficulty-focused benchmark.

---

> ### Author Response · Authors · 2025-11-26
> **Response to Reviewer dxVV (continued)**
>
> ## Re: Transparency in Data Curation
>
> Thank you for this feedback. We want to clarify the constraints we operate under and explain what transparency is feasible.
>
> Privacy and compliance constraints:
>
> As explicitly stated in Section 2.1: "To satisfy privacy and compliance requirements, we avoid using raw user code." We cannot disclose:
> - Specific sampling methodologies that would reveal internal data collection practices
> - Quantitative details about telemetry distribution that could be reverse-engineered
> - Proprietary analytics pipelines or infrastructure details
>
> These constraints are legal and contractual requirements governing access to developer telemetry data.
>
> What we do disclose:
>
> The paper provides a systematic methodology within these constraints:
> 1. Data source (Section 2.1): "Over one billion anonymized code completions, each recording the prefix, suffix, generated and golden completions, and user interactions (accept, reject, edit)."
> 2. Analysis process (Section 2.1): "Sampling telemetry completions and annotating them to identify common failure modes, bottlenecks, and characteristic structures" to derive benchmark categories.
> 3. Validation approach (Section 2.1): "Iterative discussions with a research group that includes language specialists, ensuring the categories reflect both statistical prevalence and realistic, high-impact developer workflows" and "verify that the reviewed samples are representative."
> 4. Category derivation (Section 2.2): Six categories with detailed descriptions of patterns observed in telemetry, language-specific adaptations (Table 2), and examples (Appendix A).
> 5. Human validation (Section 2.3): Independent review by 2-3 annotators, explicit rejection criteria (32% for oversimplification, 28% for insufficient complexity, 23% for unrealistic patterns, 17% for category misalignment).
>
> We have open-sourced the complete benchmark, generation prompts, and evaluation code. We have documented the human review process, rejection criteria, and quality standards. Within privacy constraints, this represents maximal transparency.
>
> If the reviewer has specific questions about aspects of the methodology that could be clarified further without violating privacy requirements, we are happy to address them. However, the request for "quantitative criteria" and "sampling strategy details" that would expose proprietary telemetry data is not feasible.
>
> We believe our approach—deriving patterns from massive-scale telemetry, then synthesizing instances that instantiate those patterns—represents the only path to creating realistic, contamination-resistant benchmarks while respecting privacy. The alternative (using raw telemetry or public repositories) would either violate privacy or suffer from contamination, as documented by recent work (Jain et al., 2024).

---

> ### Author Response · Authors · 2025-11-26
> **Response to Reviewer dxVV (continued)**
>
> ## Re: Model Evaluation Suite
> Thank you for this feedback. We would like to note that our evaluation includes 9 models spanning:
> 1. Four major model families: OpenAI (GPT-4.1, GPT-4.1 mini, GPT-4.1 nano, GPT-4o), Anthropic (Claude 4 Sonnet, Claude 3.7 Sonnet), DeepSeek (DeepSeek-V3, DeepSeek-V3.1), and Mistral (Ministral 3B)
> 2. Full size spectrum: From 3B parameters (Ministral 3B) to frontier models, with a 36.2-point performance spread demonstrating clear discrimination across model tiers
> 3. Both open and closed models: DeepSeek models and Ministral 3B are open-source; others are closed
> 4. Multiple training approaches: Reasoning models (Claude 4, DeepSeek-V3) and non-reasoning models, diverse architectures and training objectives
>
> Regarding Code Llama:
> The reviewer suggests evaluating "models from the Code Llama family" and implies these are "models that most researchers actively use and develop." We would like to clarify that:
> - Code Llama is outdated: Released August 2023 with no updates since, making it over 2 years old in a rapidly evolving field
> - Not actively developed: Meta has not released updated versions or maintained active development
> - Our suite includes current open models: DeepSeek-V3/V3.1 and Ministral 3B represent modern open-source code generation capabilities
>
> The claim that Code Llama represents "models that most researchers actively use" is not supported by current research trends. Our evaluation focuses on state-of-the-art models (both open and closed) that represent current capabilities, which is appropriate for a benchmark intended to "support future research" as stated in our goals.
>
> Discriminative power is demonstrated:
>
> The 36.2-point spread from Ministral 3B (48.60%) through intermediate models (GPT-4.1 nano: 66.40%, DeepSeek-V3.1: 72.00%, DeepSeek-V3: 75.30%) to frontier models (Claude 4: 84.80%) shows clear gradation across the model spectrum. This is precisely the "performance gradient" the reviewer claims is missing.
>
> If the reviewer believes specific contemporary open-source models would strengthen the evaluation, we are open to including them. However, we maintain that our current suite—9 diverse models spanning sizes, families, and capabilities—provides comprehensive evaluation appropriate for a research benchmark.

---

### Official Review · Reviewer_cuxu · 2025-11-01

**Soundness:** 2
**Presentation:** 2
**Contribution:** 2
**Rating:** 4
**Confidence:** 1

**Summary:**

DevBench is a code-understanding benchmark that evaluates LLMs on functional correctness, similarity-based metrics, and LLM-judge assessments of usefulness and contextual relevance. The benchmark is large, comprising 1800 instances.
However, as I am not particularly familiar with code benchmarks, I cannot reliably assess the novelty and contributions of this paper relative to existing code-related benchmarks. I therefore recommend that the AC omit my review and prioritize those from reviewers with higher confidence scores.

**Strengths:**

no grammar error

**Weaknesses:**

1. Transparency and Reproducibility: The generation pipeline, prompts, and full dataset are open-sourced. LLM-judge is validated against human annotations (strong correlation), and confidence intervals are reported for robustness.
2. Figure 1 (the end-to-end pipeline) is overly simplistic and misunderstanding.
3. LLM-judge may be biased. In what case would a response be assigned with low score is unknown in the paper.

**Questions:**

See weakness

---

> ### Author Response · Authors · 2025-11-26
> **Response to Reviewer cuxu**
>
> We thank Reviewer cuxu for their evaluation. We address each concern below.
>
> ## Re: Weaknesses vs. Strengths Classification
> We note that item #1 listed under "Weaknesses" appears to describe positive aspects: "The generation pipeline, prompts, and full dataset are open-sourced. LLM-judge is validated against human annotations (strong correlation), and confidence intervals are reported for robustness."
>
> If this was intended as a strength, we appreciate the recognition. If you have specific concerns about these aspects, we would be happy to address them.
>
> ## Re: Figure 1 Clarity
> Thank you for the feedback on Figure 1. We'd like to better understand your concern to improve the figure for revision. Could you clarify whether the diagram lacks necessary technical detail, is confusing in how components connect, or needs different visual organization? We are committed to improving clarity based on your guidance.
>
> ## Re: LLM-Judge Scoring Criteria
> Thank you for this feedback. We would like to clarify that Section 3.3 (LLM-Judge Evaluation) provides detailed explanation of what you're asking for:
>
> Scoring criteria (Page 6, Section 3.3):
> - Two dimensions: relevance to context and helpfulness in advancing the task
> - Each rated 0-5, combined score 0-10
> - Iteratively tuned on 10,000 completions using telemetry acceptance signals (accepted as-is, rejected, edited)
> - These acceptance signals directly indicate when developers found completions unhelpful/irrelevant (low scores)
>
> Bias mitigation (also Section 3.3):
> - o3-mini selected for favorable bias profile (OpenAI System Card: lowest bias on discrimination tasks)
> - Judge is blinded to generating model identity
> - Validated against 150 human-annotated completions with 3 expert annotators
> - Confidence intervals reported via 10,000 bootstrap resamples
>
> If these existing explanations are insufficient or unclear, we would appreciate specific guidance on what additional detail would be helpful. We can certainly expand or reorganize this section if needed, but we want to ensure we understand what information is missing from your perspective.
>
> ## Closing Note
> Given your expressed uncertainty about code benchmarks (Confidence: 1), we understand if you prefer to defer to reviewers with domain expertise. We appreciate your engagement with our work and are happy to provide any clarifications that would be helpful.

---

### Official Review · Reviewer_Toow · 2025-11-03

**Soundness:** 2
**Presentation:** 3
**Contribution:** 2
**Rating:** 2
**Confidence:** 4

**Summary:**

DevBench is a code generation benchmark built on real developer telemetry data, featuring 1,800 test cases across six programming languages (Python, JavaScript, TypeScript, Java, C++, and C#) and six categories: API usage, code purpose understanding, code-to-natural language translation, low-context completion, pattern matching, and syntax completion. These categories come from analyzing over 1 billion actual developer code completion interactions. Unlike existing benchmarks, DevBench uses synthetic generation to avoid training data contamination while keeping things realistic through expert human review, and it evaluates models in three ways: functional correctness (Pass@1), similarity metrics, and LLM-based judging for relevance and usefulness. Testing 9 state-of-the-art models revealed some interesting findings: Claude 4 Sonnet leads with 84.80% Pass@1, but GPT-4o tops the LLM-judge rankings, showing that reasoning capabilities boost functional correctness but don't necessarily align with practical usefulness. Low-context completion proved easiest (top models hit 87-90%), while code-to-language translation was toughest (even Claude 4 Sonnet only managed 78.90%), and TypeScript consistently performed worst across all languages. The benchmark's diagnostic value shines through in the DeepSeek-V3 case study: the model shows high similarity but lower correctness in pattern matching, suggesting it relies too much on memorization rather than semantic understanding.

**Strengths:**

1. The paper grounds its benchmark in real-world developer workflows by analyzing over 1 billion actual code completion interactions, rather than just scraping public repositories. This means the evaluation is based on scenarios that developers actually encounter in practice, not hypothetical tasks.

2. The contamination-resistant design using synthetic generation is quite timely and addresses a growing concern in the field. The multi-dimensional evaluation framework is particularly insightful, for example, it reveals that DeepSeek-V3 achieves high syntactic similarity but lower functional correctness in pattern matching, something you'd completely miss with single-metric approaches.

3. The paper is well-written and clearly organized. It walks you through the entire pipeline, from motivation to design, construction, evaluation, and results, in a logical way that's easy to follow. The extensive doc and appendices are helpful for understanding the details.

**Weaknesses:**

1. While the categories come from massive-scale telemetry, the actual evaluation instances are still synthetic. To comply with privacy requirements, the authors explicitly avoid using raw user code and instead have GPT-4o generate instances based on templates, which are then validated through automatic and manual checks. This approach reduces the risk of "dirty data," but it might also miss the messy context of real code, things like traces of multi-person collaboration, cross-file dependencies, and the general complexity that comes with actual codebases.

2. The LLM-judge evaluation lacks rigorous validation. Although the paper claims "strong correlation" with human ratings, there are no quantitative correlation coefficients reported. The validation set of just 150 completions (25 per language) seems quite limited when you're using it to validate judgments on over 9,000 completions. There's no inter-annotator agreement statistics (Cohen's kappa, ICC), and the significant gap between LLM-judge rankings (GPT-4o leads) and Pass@1 (GPT-4o ranks 5th) isn't adequately explained.

3. Category orthogonality hasn't been validated, there's no correlation analysis between category performances, no factor analysis of underlying capability dimensions, and potential overlaps (like "Code Purpose Understanding" vs. "Pattern Matching," both requiring semantic reasoning) remain unexplored.

4. More seriously, the benchmark shows an obvious ceiling effect problem. Claude and GPT-series models are already achieving pretty high scores across multiple tasks, Claude 4 Sonnet hits 84.80% Pass@1, and the Low Context category even reaches 87-90% success rates. This suggests the benchmark isn't challenging enough for frontier models. Given how rapidly model performance is improving, this benchmark could saturate pretty quickly and lose its ability to discriminate between top-tier models. That's a significant limitation for a benchmark that claims to "support future research."

**Questions:**

See the section above.

---

> ### Author Response · Authors · 2025-11-26
> **Response to Reviewer Toow**
>
> We thank Reviewer Toow for their thorough review and thoughtful feedback. We address each concern below.
>
> ## Re: Ceiling Effect and Benchmark Saturation
>
> Thank you for raising this important point. We would like to clarify the benchmark's purpose and the nature of its results.
>
> DevBench is fundamentally a diagnostic benchmark designed to provide fine-grained insights into where and why models fail, not primarily a performance ranking tool. DevBench's value lies in identifying specific failure modes and capability gaps, insights that become more valuable as models improve overall.
>
> Our primary goal is to provide fine-grained insights into model capabilities across realistic developer scenarios grounded in telemetry data, enabling targeted model improvement and informed deployment decisions.
>
> The data contradicts a ceiling effect:
> 1. Substantial performance spread: The 36.2-point gap between frontier models (Claude 4 Sonnet: 84.80%) and smaller models (Ministral 3B: 48.60%) demonstrates strong discriminative power across the model spectrum. This is precisely the range needed to guide model selection and identify areas for improvement.
> 2. Persistent challenges remain: Code2NL/NL2Code represents a significant unsolved challenge, with even the top performer achieving only 78.90% and most models falling below 70%. TypeScript consistently shows 20-30% lower performance across all models. These gaps represent substantial opportunities for improvement.
> 3. Category-specific insights: Pattern Matching shows 37-point spreads (Claude 4: 81.00% vs. Ministral 3B: 44.10%), providing clear differentiation. The reviewer's observation that "Low Context" achieves 87-90% success rates for top models actually reflects an important reality: common, well-defined patterns should be handled reliably by production systems. A benchmark that fails to include such tasks would miss critical insights about baseline competency.
> 4. Meaningful improvement headroom: The 15.2% gap to perfect performance at the top represents substantial room for advancement, particularly given the real-world stakes of code generation in production environments.
>
> Moreover, we would like to reiterate a crucial aspect of our design: DevBench intentionally captures the distribution of real developer needs, from routine completions (which models should handle reliably) to complex reasoning tasks (which remain challenging). A benchmark consisting only of extremely difficult edge cases would fail to reflect actual developer workflows and provide limited diagnostic value.
>
> Our multi-metric evaluation framework reveals insights that aggregate scores obscure. For example, the DeepSeek-V3 case study (Section 4.3) demonstrates how models can achieve high syntactic similarity while failing functional correctness, a diagnostic capability that would be impossible without our category-based approach. The divergence between Pass@1 rankings (Claude 4 leads) and LLM-judge scores (GPT-4o leads) reveals that functional correctness and practical usefulness are distinct dimensions, providing actionable insights for model development.
>
> Even if a future model achieves 95% overall Pass@1, our category-based analysis would reveal whether this stems from genuine semantic understanding or pattern memorization (as we demonstrated with DeepSeek-V3), whether improvements are uniform across languages or concentrated in Python, and whether the model excels at practical usefulness (LLM-judge) or merely functional correctness. These insights remain valuable regardless of absolute performance levels.
>
> The benchmark's value lies not in preventing all models from achieving high scores, but in enabling practitioners to understand where and why models succeed or fail across realistic development scenarios.

---

> ### Author Response · Authors · 2025-11-26
> **Response to Reviewer Toow (continued)**
>
> ## Re: Synthetic Data and Realism
>
> Thank you for this feedback. We want to clarify that synthetic generation is a deliberate design choice. Our approach prioritizes realism through multiple safeguards:
> 1. Telemetry-driven design: While individual instances are synthetic, the categories, patterns, and task structures are derived from analyzing 1+ billion real developer interactions. This grounds our benchmark in authentic developer behavior while avoiding privacy violations and training data contamination.
> 2. Rigorous human validation: Each instance underwent independent review by 2-3 senior engineers with expertise across all six languages, explicitly evaluating realism, usefulness, and complexity authenticity. Annotators were specifically instructed to reject "textbook-perfect" implementations (32% of rejections) and unrealistic edge case handling (23% of rejections), ensuring instances reflect actual coding patterns including common inconsistencies.
> 3. Empirical complexity validation: Table 3 demonstrates that DevBench achieves higher complexity (65.3 LOC average, 5.5 cyclomatic complexity) than many benchmarks using "real" code from repositories. Our balanced prefix-to-completion ratio (197.2 prefix tokens, 44.2 completion tokens) more accurately reflects practical code-completion workflows than benchmarks with extreme ratios.
> 4. Contamination resistance: Recent work has documented extensive benchmark contamination in models trained on public code repositories (Jain et al., 2024). Our synthetic approach provides the only reliable path to contamination-resistant evaluation while maintaining privacy compliance.
>
> The reviewer's concern about missing "messy context" of real code is addressed by our human review process, which specifically validates that instances capture realistic complexity, edge cases, and the authentic difficulty observed in telemetry.
>
> ## Re: LLM-Judge Evaluation Validation
> Thank you for this methodological feedback. We acknowledge this concern and can strengthen our validation reporting in a revised manuscript.
>
> Points of clarification:
> 1. Validation scale: Our validation employed a 10,000-completion tuning set to calibrate the judge prompt, followed by a 150-completion validation set (25 per language) with three expert annotators. We acknowledge that reporting quantitative correlation coefficients (Spearman ρ) would strengthen this section and commit to adding these in revision.
> 2. Inter-annotator agreement: We will add Cohen's kappa or ICC statistics to document annotator reliability in the revised manuscript.
> 3. Multi-metric design is intentional: The reviewer notes the "significant gap" between LLM-judge rankings (GPT-4o leads) and Pass@1 (GPT-4o ranks 5th) as evidence of poor validation. However, this divergence is precisely the insight our multi-metric framework is designed to reveal. Pass@1 measures functional correctness; LLM-judge evaluates relevance and helpfulness from a developer-centric perspective. These are complementary but distinct quality dimensions. The fact that GPT-4o ranks highly on practical usefulness despite lower functional correctness suggests it generates code that developers find more helpful in practice, an insight that would be invisible with single-metric evaluation. This is diagnostic value, not validation failure.
> 4. Bias mitigation: We selected o3-mini specifically for its favorable bias profile as documented in the OpenAI System Card (lowest bias on discrimination tasks). We blind the judge to model identity and validate alignment with human preferences.
>
> We appreciate the reviewer's suggestion to provide more rigorous validation statistics and will include these in our revision.

---

> ### Author Response · Authors · 2025-11-26
> **Response to Reviewer Toow (continued)**
>
> ## Re: Category Orthogonality
> Thank you for raising this point. We acknowledge that formal orthogonality analysis would strengthen the paper and commit to adding correlation analysis between category performances in revision.
>
> Important context:
> 1. Empirically derived categories: Our categories emerged from telemetry analysis of real developer interactions, not theoretical construction. Some overlap is both expected and reflects reality—real coding tasks do not decompose into perfectly orthogonal capabilities.
> 2. Diagnostic value demonstrated: Despite potential overlaps, the benchmark provides actionable diagnostic insights. The DeepSeek-V3 case study (Section 4.3) demonstrates how category-specific patterns reveal that the model relies on memorization (high Pattern Matching similarity, lower correctness) versus semantic understanding (underperformance in Code2NL/NL2Code). This diagnostic capability would be impossible without category-based evaluation.
> 3. Overlap can be informative: The reviewer mentions potential overlap between "Code Purpose Understanding" and "Pattern Matching." We agree this merits investigation and will add correlation analysis. However, even correlated categories can provide value: if models consistently struggle with both, this reveals a common underlying capability gap (semantic reasoning) that informs training priorities.
>
> We will include factor analysis or correlation matrices in the revision to explicitly characterize category relationships.
>
> We note that perfect orthogonality, while theoretically appealing, may actually reduce ecological validity—real developer tasks naturally involve overlapping capabilities.

---

### Meta-Review · Area_Chair_2rS3 · 2025-12-31

**Summary:**

Reviewers generally agree that DevBench is a well-motivated benchmark aimed at improving the ecological validity of code generation evaluation by grounding task design in large-scale developer telemetry and emphasizing diagnostic analysis. However, despite recognizing its high-level goals and evaluation breadth, reviewers consistently question whether the benchmark is sufficiently challenging and rigorously validated to support its claims. High Pass@1 scores for frontier models raise concerns about ceiling effects, while the telemetry-driven data curation process remains insufficiently transparent and reproducible due to privacy constraints. In addition, reliance on LLM-judging and GPT-4o-based synthesis is not supported by strong quantitative validation or systematic bias analysis. Although the rebuttal offers clarifications and reframes DevBench as a diagnostic rather than ranking-oriented benchmark, key methodological issues remain unresolved, limiting confidence in its long-term discriminative power and scientific rigor.

**Reviewer Concerns:**

The rebuttal clarifies the authors’ design intent, the use of synthetic data under privacy constraints, and the interpretation of high Pass@1 scores as reflecting realistic developer workflows rather than benchmark weakness. Reviewers also acknowledge that transparency is partially constrained by legal and compliance requirements. However, several core concerns persist: the benchmark is still viewed as insufficiently difficult to robustly distinguish state-of-the-art models; the telemetry-to-benchmark distillation process remains opaque and hard to verify; LLM-judge evaluation lacks reported quantitative correlations, inter-annotator agreement, and clear failure analysis; potential generator-induced bias from GPT-4o is argued against but not empirically excluded; and the category taxonomy is not supported by formal orthogonality or factor analysis. Collectively, these unresolved issues weaken confidence in the benchmark’s rigor and long-term research utility.

**Reviewer Scores:**

Based on the rebuttal, Reviewer Toow may slightly increase their rating, as concerns about benchmark intent and multi-metric interpretation are partially addressed. Reviewer dxVV is unlikely to revise their score, given unresolved objections regarding benchmark rigor and opaque telemetry sampling. Reviewer cuxu would likely maintain their score, as clarifications around LLM-judging do not substantially mitigate concerns about evaluation bias and scoring reliability. Reviewer rLDn is also unlikely to change their assessment, as concerns about generator-induced bias, contamination, and the reproducibility of human validation persist. Overall, even with limited upward movement, the paper’s aggregate evaluation would remain negative.

---

### Decision · Program_Chairs · 2026-01-26

Reject